# ENHANCING MOLECULAR PROPERTY PREDICTIONS BY LEARNING FROM BOND MODELLING AND INTERACTIONS

**Yunqing Liu**[1]    **Yi Zhou**[1]    **Wenqi Fan**[1,2]*

[1]Department of Computing (COMP), The Hong Kong Polytechnic University
[2]Department of Management and Marketing (MM), The Hong Kong Polytechnic University
`{yunqing617.liu, echo-yi.zhou}@connect.polyu.hk`
`wenqifan03@gmail.com`

## ABSTRACT

Molecule representation learning is crucial for understanding and predicting molecular properties. However, conventional atom-centric models, which treat chemical bonds merely as pairwise interactions, often overlook complex bond-level phenomena like resonance and stereoselectivity. This oversight limits their predictive accuracy for nuanced chemical behaviors. To address this limitation, we introduce **DeMol**, a dual-graph framework whose architecture is motivated by a rigorous information-theoretic analysis demonstrating the information gain from a bond-centric perspective. DeMol explicitly models molecules through parallel atom-centric and bond-centric channels. These are synergistically fused by multi-scale Double-Helix Blocks designed to learn intricate atom-atom, atom-bond, and bond-bond interactions. The framework's geometric consistency is further enhanced by a regularization term based on covalent radii to enforce chemically plausible structures. Comprehensive evaluations on diverse benchmarks, including PCQM4Mv2, OC20 IS2RE, QM9, and MoleculeNet, show that DeMol establishes a new state-of-the-art, outperforming existing methods. These results confirm the superiority of explicitly modelling bond information and interactions, paving the way for more robust and accurate molecular machine learning.

## 1  INTRODUCTION

Molecules, the fundamental building blocks of matter, govern the properties and behaviours of everything from simple compounds to complex biological systems (Fan et al., 2025; Liu et al., 2025b; Zhou et al., 2025). Their intricate three-dimensional architectures and chemical interactions determine the functionality of drugs, materials, catalysts, and biomolecules, making molecular understanding indispensable in fields such as medicine (Li et al., 2024b; Zhao et al., 2024), energy (Wang et al., 2025a; Hu et al., 2023), and environmental science (Liu et al., 2025a; 2023). De-

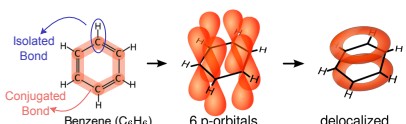

Figure 1: **An example of the different bonds within benzene ($C_6H_6$).** Formed by $p$-orbital overlap, the delocalized $\pi$ system creates electron density above/below the ring plane, providing stability and unique reactivity.

ciphering these relationships requires not only experimental techniques but also computational approaches that can model and predict molecular behaviour at scale, which is a challenge that has driven the rise of molecular representation learning.

Traditional methods for molecular analysis rely on handcrafted descriptors (e.g., molecular fingerprints) or physics-based simulations, which are labour-intensive and often limited in capturing complex structural dependencies (Duvenaud et al., 2015; Collins & Bettens, 2015; Li et al., 2024a;c). With the advent of graph neural networks (GNNs) in AI techniques (Fan et al., 2022; Zhang et al.,

---

*Corresponding author: Wenqi Fan, Department of Computing & Department of Management and Marketing (MM), The Hong Kong Polytechnic University

2024; Ma et al., 2026), molecules are now widely described as graphs, where atoms (i.e., *nodes*) and bonds (i.e., *edges*) form interconnected networks, enabling end-to-end learning of task-specific embeddings (Luo et al., 2022; Lu et al., 2023). Recent methods have shifted toward modelling molecules as 3D geometric graphs, leveraging spatial coordinates of atoms to capture directional bonding patterns, steric effects, and non-covalent interactions critical for accurate property prediction (Liao & Smidt, 2022; Lu et al., 2023; Hussain et al., 2024).

However, most existing methods treat atoms as primary entities, neglecting the rich information embedded in chemical bonds themselves (Ying et al., 2021; Xia et al., 2023; Luo et al., 2022). Bonds are not only pairwise interactions, but also carry attributes like bond order, length, and hybridisation states that directly influence molecular reactivity and stability (Evans, 2001). For instance, as Figure 1 shown, the alternating single and double bonds on the benzene ring are not isolated but form a delocalized $\pi$-electron system through inter-bond resonance. This collective behaviour cannot be described by pairwise atomic interactions alone. Another critical yet underexplored aspect of molecular modelling is the explicit capture of interactions between chemical bonds. While bonds are typically treated as independent edges in graph-based models, real-world molecules exhibit non-additive bond interactions that govern phenomena such as stereoselectivity and spatial cooperativity (Ding et al., 2019; Weng et al., 2021). For example, as illustrated in Figure 2, cisplatin is an anticancer drug that belongs to the group of cell cycle non-specific drugs. It is therapeutically effective against sarcomas, malignant epithelial tumours, lymphomas, and germ cell tumours. To be specific, *cisplatin* is a platinum-containing compound in which two ammonia ligands and two chloride ions bind to a central platinum atom in a *cis* configuration. This spatial arrangement enables cisplatin to crosslink DNA strands, disrupting replication and inducing apoptosis in cancer cells (Dong et al., 2019). In contrast, *transplatin*, the stereoisomer of cisplatin, possesses the same atomic composition but features ligands in a *trans* configuration, rendering it pharmacologically ineffective. Therefore, these striking differences arise not from changes in individual bond attributes (e.g., bond length or hybridisation) but from the collective orientation of bonds relativ

To eliminate these limitations, we propose to model the molecule bond interactions into the molecule representation learning, thereby integrating the bond and interaction information for molecular property prediction. We first establish the theoretical necessity of bond-centric graph in capturing bond-centric attributes and interactions in molecule representation learning, as shown in Section 3.1. We propose a novel **D**ual-graph **e**nhanced **M**ulti-scale interaction framework for **Mol**ecule representation learning, **DeMol**, which is a hierarchical multi-scale framework that explicitly models both atoms and bonds through dual-graph representations and geometric constraints, including atom-centric channel and bond-centric channel and double-helix blocks. As shown in Figure 3, two channels encode a molecule into the atom-centric graph and bond-centric graph, respectively. The double-helix blocks then facilitate information exchange and fusion between these two channels at multiple scales, enabling the model to capture intricate atom-atom, atom-bond, and bond-bond interactions for a

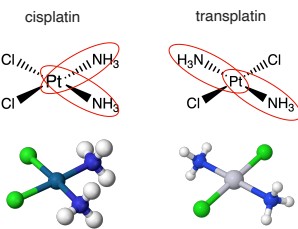

Figure 2: An example of bond-bond interactions in molecules that affect properties. **Left** is the anticancer drug cisplatin, where two ammonia ligands and two chloride ions bind to a central platinum atom in a *cis* configuration. **Right** is transplatin, which possesses the same atomic composition but features ligands in a *trans* configuration, rendering it pharmacologically ineffective.

comprehensive understanding of the molecule. To ensure geometric consistency, we also introduce bond prediction based on covalent radii as the regularisation term, as shown in Algorithm 1, which enforces chemically plausible structures by penalising deviations from expected bonding distances. Finally, we evaluate model performance on various molecular property prediction tasks. DeMol outperforms the state-of-the-art methods on PCQM4Mv2, OC20 IS2RE, QM9, and MoleculeNet datasets, which verifies the performance superiority of DeMol.

## 2  RELATED WORK

**Graph-based Molecule Representation Learning.** Molecule representation learning aims to encode molecular structures into continuous vector spaces, capturing their chemical knowledge to facilitate downstream tasks (Guo et al., 2022). As one of the most representative AI techniques (Xu et al., 2025; Ning et al., 2025; Qu et al., 2026; Wang et al., 2025b; Fan et al., 2024; Ning et al., 2024; Xiao

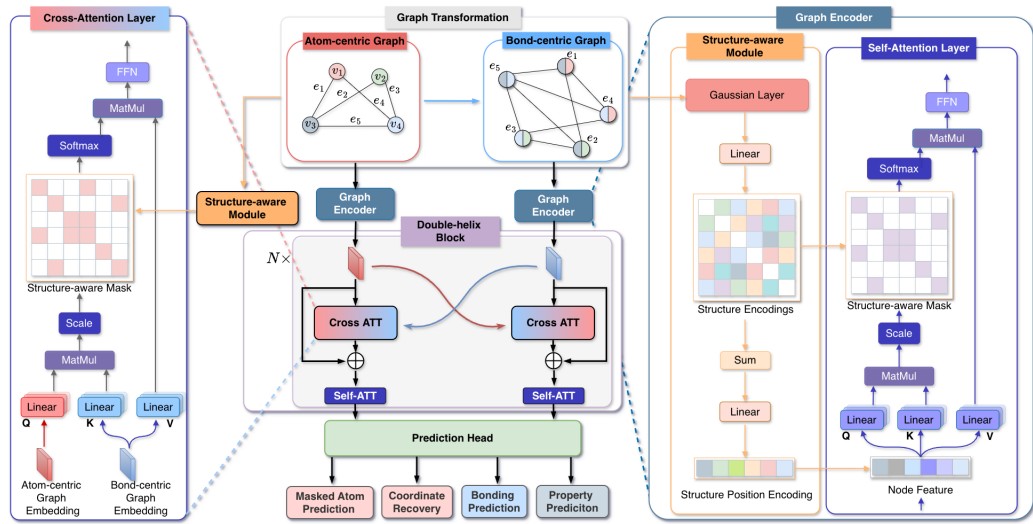

Figure 3: DeMol integrates atom-centric and bond-centric channels via dual-graph representations. Cross-level (atom-bond) interactions are enforced through double-helix blocks, ensuring geometric consistency.

et al., 2026; Zhou et al., 2026), GNNs and graph transformers encode atomic topological information into molecule representations via message-passing (Gasteiger et al., 2020; 2021; Ying et al., 2021; Rampášek et al., 2022; Wang et al., 2020; Fan et al., 2019; 2020). Various geometry-aware GNNs and equivariant neural networks additionally learns molecular spatial geometry knowledge (Liu et al., 2022b; Liao & Smidt, 2022; Zhou et al., 2023; Wang et al., 2023b). Furthermore, some unified 2D/3D molecular representation learning frameworks have also been proposed to make the best of topological and geometric features integrally (Liu et al., 2021; Luo et al., 2022; Lu et al., 2023; Stärk et al., 2022).

**Molecular Bond Modelling.** In recent years, advances in 3D molecular representation learning have implicitly incorporated bond-related information to enhance atomic representations. DimeNet (Gasteiger et al., 2020) models interatomic distances and angles as structural constraints. GemNet (Gasteiger et al., 2021) involves dihedral angles in the geometric representation and message-passing. Transformer-M (Luo et al., 2022) integrates 3D distance encoding with the attention mechanism. Meanwhile, there are individual pioneering studies that attempt to focus more on chemical bonds. LEMON (Chen et al., 2025) employs line graphs as a contrast to molecule graphs, and GEM (Fang et al., 2022) encodes molecular geometries by modelling on bond-angle graphs. ALIGNN (Choudhary & DeCost, 2021) uses a line graph, treating it as a route for message passing. ESA (Buterez et al., 2025) introduces an end-to-end attention architecture that treats graphs as sets of edges to explicitly learn edge representations. These approaches have demonstrated the value of bond features, whether as auxiliary information or independent modeling elements. However, despite the achieved success, existing molecule modelling methods still suffer from unbalanced atom/bond modeling and geometric simplification. A coordinated framework explicitly learning dynamic atom-atom, atom-bond, and bond-bond interactions remains required for understanding more complete molecular semantics.

## 3 METHODOLOGY

We first establish the theoretical necessity of a bond-centric graph in capturing bond-centric attributes and interactions in molecular representation learning. This theoretical foundation motivates our DeMol framework, which explicitly models atom–atom, atom–bond, and bond–bond interactions to achieve a more comprehensive understanding of molecular structures.

**Problem Formulation and Notations.** As shown in Figure 3, a molecule $\mathcal{M}$ is represented by two graphs. For the atom-centric graph $\mathcal{G} = (\mathcal{V}, \mathcal{E})$, node $v_i \in \mathcal{V}$ represents atoms with feature $\mathbf{x}_i \in \mathbb{R}^{d_a}$, and edge $e_{ij} \in \mathcal{E}$ encode bond attributes $\mathbf{e}_{ij} \in \mathbb{R}^{d_b}$. For bond-centric graph $\mathcal{L}(\mathcal{G}) = (\mathcal{V}', \mathcal{E}')$, node $v'_{ij} \in \mathcal{V}'$ represents bonds from $\mathcal{G}$, with features derived from $\mathbf{e}_{ij}$, and edge $e'_{(ij)(jk)} \in \mathcal{E}'$ connect bonds sharing a common atom $j$, encoding angular relationships $\theta_{ijk}$ and torsional angles $\phi_{ijkl}$. We define the dual graph representation learning objective as mapping $(\mathcal{G}, \mathcal{L}(\mathcal{G})) \to (\mathbf{H}^{(a)}, \mathbf{H}^{(b)})$, where $\mathbf{H}^{(a)} \in \mathbb{R}^{N \times d_a}$ is the atom-level embeddings derived from $\mathcal{G}$ and $\mathbf{H}^{(b)} \in \mathbb{R}^{M \times d_b}$ is the

bond-level embeddings derived from $\mathcal{L}(\mathcal{G})$. Our goal is to maximise mutual information between representations and molecular properties while preserving geometric consistency.

## 3.1 THEORETICAL MOTIVATION AND RATIONALE

**Proposition 1 (Information Gain from Edge Adjacency Patterns).** *For non-trivial graphs $\mathcal{G}$, the entropy of $\mathcal{L}(\mathcal{G})$ satisfies:*

$$\mathbf{H}(\mathcal{L}(\mathcal{G})) = \mathbf{H}(\mathcal{G}) + \underbrace{\mathbf{H}(\mathcal{E}'|\mathcal{E})}_{\Delta I_{\text{edge-structure}}} \quad \text{with} \quad \Delta I_{\text{edge-structure}} > 0. \tag{1}$$

This demonstrates that while the bond-centric graph is derived from the original graph, it encodes unique structural information absent in the original. The justification can be found in Appendix A.1. To leverage this distinct source of information, we designed DeMol with two parallel processing streams. Instead of treating bonds merely as edges in an atom-centric graph, we create a dedicated bond-centric channel that processes $\mathcal{L}(\mathcal{G})$ as a first-class entity. This dual-channel design ensures that the unique structural information inherent to both atoms and bonds is independently captured and refined from the outset.

**Proposition 2 (Mutual Information Decomposition).** *For any molecular graph $\mathcal{G}$ and its dual-graph representation $(\mathbf{H}^{(a)}, \mathbf{H}^{(b)})$, the mutual information satisfies:*

$$I(\mathcal{G}, \mathcal{L}(\mathcal{G}); \mathbf{H}^{(a)}, \mathbf{H}^{(b)}) = \underbrace{I(\mathcal{G}; \mathbf{H}^{(a)})}_{\text{Atom-Centric Information}} + \underbrace{I(\mathcal{L}(\mathcal{G}); \mathbf{H}^{(b)}|\mathbf{H}^{(a)})}_{\text{Bond-Centric Information}} + \underbrace{I(\mathcal{G}; \mathbf{H}^{(b)}|\mathcal{L}(\mathcal{G}), \mathbf{H}^{(a)})}_{\text{Residual Atomic Dependency}}. \tag{2}$$

This decomposition demonstrates that dual-graph learning retains strictly more information than single-graph approaches when $I(\mathcal{L}(\mathcal{G}); \mathbf{H}^{(b)}|\mathbf{H}^{(a)}) > 0$ and $I(\mathcal{G}; \mathbf{H}^{(b)}|\mathcal{L}(\mathcal{G}), \mathbf{H}^{(a)}) > 0$. The justification can be found in Appendix A.2. Proposition 2 decomposes the total mutual information, revealing that a dual-graph representation can capture strictly more information than either graph alone by combining atom-centric information, bond-centric information, and their residual dependencies. This highlights that the ultimate predictive power lies not in the separate representations, but in their effective fusion. A simple concatenation of features would fail to capture the complex cross-dependencies between atoms and bonds. To achieve a true synergistic fusion, we introduce the Double-Helix Blocks. This mechanism employs a bidirectional cross-attention module that facilitates a dynamic interaction between the atom and bond representations at multiple scales. This allows the model to explicitly learn and integrate complex atom-atom, bond-bond, and atom-bond interactions, directly optimising the information fusion highlighted as essential by our analysis.

**Proposition 3 (Geometric Information Gain).** *Let $\theta_{ijk}$ denote bond angles and $\varphi_{ijkl}$ denote dihedral angles. If $\mathcal{L}(\mathcal{G})$ encodes angular relationships through $\mathcal{E}'$, the bond modelling gain satisfies:*

$$\Delta I = I(\mathcal{L}(\mathcal{G}); \mathbf{H}^{(b)}|\mathcal{G}; \mathbf{H}^{(a)}) \propto \mathbb{E}_{i,j,k,l} \left[ \log \frac{p(\theta_{ijk}, \varphi_{ijkl}|\mathbf{h_i}^{(a)}, \mathbf{h_j}^{(a)}, \mathbf{h_k}^{(a)}, \mathbf{h_l}^{(a)})}{p(\theta_{ijk}, \varphi_{ijkl})} \right]. \tag{3}$$

Proposition 3 reveals that the bond-centric graph is the natural domain for representing complex geometric relationships like bond angles ($\theta_{ijk}$) and dihedral angles ($\varphi_{ijkl}$), which are only implicitly captured in an atom-centric view. The structure of $\mathcal{L}(\mathcal{G})$, where bonds are nodes, makes relationships between adjacent bonds (angles) and pairs of bonds (dihedrals) explicit. The justification can be found in Appendix A.3. To directly harness this geometric advantage, we introduce a torsion encoding matrix ($\Phi_b^{tors}$) specifically within the bond-centric channel. By incorporating bond angles and inter-bond dihedral angles as an attention bias in this channel, we inject critical 3D information where it is most naturally represented. This targeted design choice, guided by our theoretical insight, ensures that the model can more effectively learn from the molecule's full geometric conformation.

**Proposition 4 (Dual-Graph Information Bottleneck).** *For molecular property prediction $\mathbf{Y}$, the optimal encoder $\mathbf{\Phi}$ minimizes:*

$$\min_{\mathbf{\Phi}} \left[ I(\mathcal{G}, \mathcal{L}(\mathcal{G}); \mathbf{H}^{(a)}, \mathbf{H}^{(b)}) - \beta I(\mathbf{H}^{(a)}, \mathbf{H}^{(b)}; \mathbf{Y}) \right]. \tag{4}$$

Proposition 4 applies the Information Bottleneck principle to our dual-graph setup. It suggests that the optimal model must learn to compress the rich information from both graphs into a minimal representation that is maximally predictive of the target property. The richer input from a dual-graph system provides a tighter bound on this optimisation, but also increases the risk of learning from spurious or non-physical correlations. This motivates our end-to-end learning approach, which culminates in a final Prediction Head that leverages the fused atom-bond representations. Furthermore, to ensure the model achieves an effective information-bottleneck trade-off, we introduce two key regularisation and efficiency mechanisms: *Bond Prediction based on Covalent Radii* and *Structure-aware Mask*.

Our theoretical analysis establishes that dual-graph representations (atom-centric graph and bond-centric graph) retain strictly more information than single-graph approaches (Proposition 1 and 2) and improve geometric consistency by explicitly encoding angular relationships (Proposition 3). The optimal encoder balances information compression with task-relevant prediction (Proposition 4).

## 3.2 PROPOSED FRAMEWORK

The previous theoretical foundation motivates our proposed dual-graph enhanced multi-scale interaction framework (DeMol), as shown in Figure 3. To model both $\mathcal{G}$ and $\mathcal{L}(\mathcal{G})$, DeMol employs dual encoding channels that activate atom-centric and bond-centric representations in parallel.

**Atom-centric Channel on $\mathcal{G}$.** This channel processes the atom-centric graph $\mathcal{G}$ to learn atom embeddings $\{\mathbf{h_i}^{(a)}\}$. Firstly, raw atom feature $\mathbf{x}_i$ are embedded into $\mathbf{h_i}^{(a,0)} \in \mathbb{R}^{d_a}$. Similar to previous works, we use structure encoding to encode the 3D spatial and 2D graph positional information (Zhou et al., 2023; Luo et al., 2022). As for 3D structure encodings, we encode the Euclidean distance to reflect the spatial relation between any pair of atoms in the 3D space. For each atom pair $(i, j)$, we first process their Euclidean distance with Gaussian Basis Kernel function (Scholkopf et al., 1997)

$$\phi_{(i,j)}^k = -\frac{1}{\sqrt{2\pi}|\sigma^k|} \exp(-\frac{1}{2}(\frac{\alpha_{(i,j)}||\mathbf{r_i} - \mathbf{r_j}|| + \beta_{(i,j)} - \mu^k}{|\sigma^k|})^2), k = \{1, 2, \dots, K\}, \tag{5}$$

where $K$ is the number of Gaussian Basis kernels. The input 3D coordinate of the $i$-th atom is represented by $\mathbf{r}_i \in \mathbb{R}^3$. $\alpha_{(i,j)}$ and $\beta_{(i,j)}$ are learnable scalars indexed by the pair of atom types, and $\mu^k, \sigma^k$ are predefined constants. Specifically, $\mu^k = w \times (k-1)/K$ and $\sigma^k = w/K$, where the width $w$ is a hyper-parameter. Then, the 3D distance encoding can be calculated as $\Phi_{(i,j)}^{dist} = GELU(\boldsymbol{\phi}_{(i,j)} \boldsymbol{W}_D^1) \boldsymbol{W}_D^2$, where $\boldsymbol{\phi}_{(i,j)} = [\phi_{(i,j)}^1; \dots; \phi_{(i,j)}^K]^\top$; $\boldsymbol{W}_D^1 \in \mathbb{R}^{K \times K}$, $\boldsymbol{W}_D^2 \in \mathbb{R}^{K \times 1}$ are learnable parameters. Denote $\Phi^{dist}$ as the matrix form of the 3D distance encoding, whose shape is $N \times N$. As for 2D graph structure encodings, we encode the shortest path distance (SPD) between two atoms to reflect their spatial relation. Let $\Phi_{ij}^{SPD}$ denotes the SPD encoding between atom $i$ and $j$. For most molecules, there exists only one distinct shortest path between any two atoms. Denote the edges in the shortest path from $i$ to $j$ as $\mathbf{SP}_{ij} = (s_1, s_2, \dots, e_N)$ and $\Phi^{SPD} \in \mathbb{R}^{N \times N}$ as the matrix from the SPD encoding. Combined above, the structure encoding is denoted as $\Phi = \Phi^{dist} + \Phi^{SPD}$.

For each atom $i$, the update process contains a self-attention and a feed-forward network (FFN) layer. The attention weights and the atom representations are updated as

$$\alpha_{ij}^{(l)} = \text{Softmax}_j \left( \frac{(\mathbf{W}_q^{(l)} \mathbf{h}_i^{(a,l)})^\top (\mathbf{W}_k^{(l)} \mathbf{h}_j^{(a,l)})}{\sqrt{d_k}} + \Phi^{(l)} \right), \mathbf{h}_i^{(a,l+1)} = \text{FFN}(\mathbf{h}_i^{(a,l)} + \sum_{j \in \boldsymbol{\mathcal{V}(i)}} \alpha_{ij}^{(l)} \mathbf{W}_v^{(l)} \mathbf{h}_j^{(a,l)}), \tag{6}$$

where $\mathbf{W}_q, \mathbf{W}_k, \mathbf{W}_v \in \mathbb{R}^{d_a \times d_h}$ are learnable parameters.

**Bond-centric Channel on $\mathcal{L}(\mathcal{G})$.** Similar to $\mathcal{G}$, we process the bond-centric graph $\mathcal{L}(\mathcal{G})$ to learn bond embeddings $\{\mathbf{h}_{ij}^{(b)}\}$. We first encode bond feature $\mathbf{e}_{ij}$ as $\mathbf{h}_{i,j}^{(b,0)} \in \mathbb{R}^{d_b}$. Similarly, we use

3D spatial and 2D graph positional information to obtain structure encodings. Denote $\Phi_b^{SPD}$ and $\Phi_b^{distance}$ as the SPD encoding and 3D distance encoding between bond $e_{ij}$ and $e_{jk}$ respectively, which are shape $M \times M$. Specifically, we use the coordinates of the midpoint of two atoms as the coordinate of the bond $\mathbf{r}_{ij} = \frac{1}{2}(\mathbf{r}_i + \mathbf{r}_j)$. In addition to this, we propose a new 3D structure positional encoding for bond-centric, named torsion encoding $\Phi_b^{tors}$, which denoted as the shape $M \times M$ matrix of bond angles $\theta_{ijk}$ and dihedral angles $\varphi_{ijkl}$. If two bonds share one atom, the bond angle can be computed as $\cos(\theta_{ijk}) = \frac{\mathbf{r}_{ij} \cdot \mathbf{r}_{jk}}{\|\mathbf{r}_{ij}\|\|\mathbf{r}_{jk}\|}$. Otherwise, the dihedral angles can be computed as $\cos(\varphi_{ijkl}) = \frac{\mathbf{r}_{ij} \cdot \mathbf{r}_{kl}}{\|\mathbf{r}_{ij}\|\|\mathbf{r}_{kl}\|}$. Similar to the atom-centric channel, we process torsion encoding with a Gaussian Basis Kernel function. Combined above, the structure encoding is denoted as $\Phi_b = \Phi_b^{dist} + \Phi_b^{SPD} + \Phi_b^{tors}$. For each bond $ij$, the attention weights and the bond representations are updated as

$$\beta_{(ij)(jk)}^{(l)} = \text{Softmax}_{(jk)}\left(\frac{(\mathbf{W}_q^{(l)}\mathbf{h}_{ij}^{(b,l)})^\top(\mathbf{W}_k^{(l)}\mathbf{h}_{jk}^{(b,l)})}{\sqrt{d_k}} + \Phi_b^{(l)}\right), \tag{7}$$

$$\mathbf{h}_{ij}^{(b,l+1)} = \text{FFN}(\mathbf{h}_{ij}^{(b,l)} + \sum_{k \in \mathcal{V}'(ij)} \beta_{(ij)(jk)}^{(l)} \mathbf{W}_v^{(l)} \mathbf{h}_{jk}^{(b,l)}), \tag{8}$$

where $\mathbf{W}_q, \mathbf{W}_k, \mathbf{W}_v \in \mathbb{R}^{d_b \times d_h}$ are learnable parameters.

**Cross-Graph Interaction via Double-Helix Blocks.** To align atom and bond representations, we introduce Double-Helix Blocks that enforce multi-scale consistency through cross-attention. For atom $i$, cross-attention over its incident bonds $\{ij\}$:

$$\gamma_{i,ij}^{(l)} = \text{Softmax}_{ij}\left(\frac{(\mathbf{W}_q^{(l)}\mathbf{h}_i^{(a,l)})^\top(\mathbf{W}_k^{(l)}\mathbf{h}_{ij}^{(b,l)})}{\sqrt{d_k}} + \Phi^{(l)}\right), \mathbf{h}_i^{(a,l+1)} = \text{FFN}(\mathbf{h}_i^{(a,l)} + \sum_{k \in \mathcal{V}(i)} \gamma_{i,ij}^{(l)} \mathbf{W}_v^{(l)} \mathbf{h}_{ij}^{(b,l)}). \tag{9}$$

For bond $ij$, cross-attention over its atoms $i, j$:

$$\delta_{ij,i}^{(l)} = \text{Softmax}_i\left(\frac{(\mathbf{W}_q^{(l)}\mathbf{h}_{ij}^{(b,l)})^\top(\mathbf{W}_k^{(l)}\mathbf{h}_i^{(a,l)})}{\sqrt{d_k}} + \Phi_b^{(l)}\right), \mathbf{h}_{ij}^{(b,l+1)} = \text{FFN}(\mathbf{h}_{ij}^{(b,l)} + \sum_{k \in \{i,j\}} \delta_{ij,k}^{(l)} \mathbf{W}_v^{(l)} \mathbf{h}_k^{(a,l)}). \tag{10}$$

**Bond Prediction based on Covalent Radii.** To ensure geometric consistency in molecular representation learning, DeMol explicitly models atom and bond interactions through a bond prediction module grounded in covalent radii constraints. Given a molecule $\mathcal{M}$ with atoms $\{a_i\}_{i=1}^N$, their 3D coordinates $\{\vec{p_i} = (x_i, y_i, z_i)\}_{i=1}^N$, and a dictionary of covalent radii $R_{cov}$, our algorithm (Appendix B Algorithm 1) predicts bonds by comparing interatomic distance to a threshold derived from covalent radii. According to (Lu & Chen, 2012), we set the bond threshold factor $\alpha$ to 1.15. The Bond Prediction based on Covalent Radii module acts as a regularisation term, penalising the model for generating geometrically inconsistent structures and ensuring the learned representations adhere to fundamental chemical principles.

**Structure-aware Mask.** Due to the introduction of bond-centric channel on $\mathcal{L}(\mathcal{G})$ and cross-graph interaction blocks, the time complexity of the model could increase, see the Appendix C for a detailed time complexity analysis. Whereupon, we introduce a structure-aware mask to mitigate this through sparse attention masks derived from a chemical valency rule, where bond lengths between atoms of covalent molecules are generally less than 3 Å (Nikolaienko et al., 2019). Considering that there are still some weak interaction forces between the atoms, we take the interatomic distance less than 5 Å as the basis for masking or not. Unlike using the adjacency matrix as a mask, the structure-aware mask both preserves the adjacent interconnectivity of the atoms and captures the potential long-range forces between the atoms. As for the bond structure-aware mask, we only consider the adjacency and conjugacy relations to enhance the ability of capturing the conjugacy relations between edges (Nikolaienko et al., 2019).

## 4 EXPERIMENTS

In this section, we empirically study the performance of DeMol. First, we pre-train our model on the training set of PCQM4Mv2, which is derived from OGB Large-Scale Challenge (Hu et al., 2021). Then, we evaluate our model in various downstream tasks through fine-tuning, including the PCQM4Mv2, Open Catalyst 2020 IS2RE, QM9, and MoleculeNet. Finally, we conduct a series of experiments to investigate the key designs of our model for ablation studies.

**Pretraining Dataset.** The pertaining dataset PCQM4Mv2 (Hu et al., 2021) is designed to facilitate the development and evaluation of machine learning models for predicting quantum chemical (QC) properties of molecules, specifically the target property known as the HOMO-LUMO gap. This property represents the difference between the energies of the highest occupied molecular orbital (HOMO) and the lowest unoccupied molecular orbital (LUMO). The dataset, consisting of 3.37 million molecules represented by SMILES notations, offers HOMO-LUMO gap labels for the training and validation sets. Furthermore, the training set encompasses the DFT equilibrium conformation, which is not included in the validation sets. On this benchmark, models are required to utilize SMILES notation, without DFT equilibrium conformation, to predict the HOMO-LUMO gap during inference.

**PCQM4Mv2.** After pre-training, we evaluate DeMol on the PCQM4Mv2 validation set. The task involves predicting the HOMO-LUMO energy gap, with Mean Absolute Error (MAE) as the evaluation metric. Since our pre-training objectives already include the HOMO-LUMO gap prediction, we evaluate the model without additional fine-tuning. For comparison, we select diverse baselines including both graph neural networks (GNNs) and Transformer variants. Detailed descriptions of settings and baselines are presented in Appendix D.2.

Table 1: Performance on PCQM4MV2 validation set (Hu et al., 2021). Bold values indicate the best performance.

| Model | # param. | MAE ($\downarrow$) |
|---|---|---|
| MLP-Fingerprint | 16.1M | 0.1735 |
| GCN | 2.0M | 0.1379 |
| GIN | 3.8M | 0.1195 |
| GINE$_{VN}$ | 13.2M | 0.1167 |
| GCN$_{VN}$ | 4.9M | 0.1153 |
| GIN$_{VN}$ | 6.7M | 0.1083 |
| DeeperGCN$_{VN}$ | 25.5M | 0.1021 |
| GraphGPS$_{SMALL}$ | 6.2M | 0.0938 |
| TokenGT | 48.5M | 0.0910 |
| GRPE$_{BASE}$ | 46.2M | 0.0890 |
| EGT | 89.3M | 0.0869 |
| GRPE$_{LARGE}$ | 46.2M | 0.0867 |
| Graphormer | 47.1M | 0.0864 |
| GraphGPS$_{BASE}$ | 19.4M | 0.0858 |
| GraphGPS$_{DEEP}$ | 13.8M | 0.0852 |
| GEM-2 | 32.1M | 0.0793 |
| GPS++ | 44.3M | 0.0778 |
| Transformer-M | 69M | 0.0772 |
| Unimol+ | 77M | 0.0693 |
| TGT-At | 203M | 0.0671 |
| DeMol | 186M | **0.0603** |

Table 1 presents the comparative results, where DeMol establishes a new state-of-the-art with an MAE of 0.0603 eV, reflecting a substantial leap in capturing molecular graph information with higher precision. Concretely, DeMol represents a 0.0068 eV (10.1%) improvement over the previous best model (TGT-At, 0.0671 eV) and significantly outperforms other approaches like Transformer-M (0.0772 eV) and GraphGPS$_{DEEP}$ (0.0852 eV). Notably, while prior top-performing models relied on extensive ensembles (e.g., GPS++'s 112-model ensemble), DeMol achieves superior performance using only a single model, demonstrating its inherent robustness and generalisation ability.

**Open Catalyst 2020 IS2RE.** In the Open Catalyst 2020 Challenge (Chanussot et al., 2021), machine learning approaches are required to predict molecular adsorption energies on catalyst surfaces. Specifically, we focus on the IS2RE (Initial Structure to Relaxed Energy) task, which comprises approximately 460K samples. In this task, the model is given an initial DFT structure of both the crystal and the adsorbate. These components interact during relaxation, and the model's goal is to predict the system's final relaxed energy. Moreover, while DFT equilibrium confirmations are provided for training, they cannot be used during inference. Appendix D.3 contains detailed descriptions of settings and baselines.

Table 2 compares model performance on the OC20 IS2RE validation set using two key metrics: Energy Mean Absolute Error (MAE) in electron volts (eV) and the percentage of Energies within a Threshold (EwT), where lower MAE and higher EwT indicate better performance. DeMol achieves state-of-the-art results, surpassing all baselines in both metrics. Specifically, DeMol obtains the lowest average energy MAE (0.3879 eV), representing 5.1% and 3.7% improvements over Unimol+ (0.4088 eV) and TGT-At (0.4030 eV), respectively. In terms of EwT, DeMol demonstrates a significant lead with an average 9.23%, compared to 8.61% for Unimol+ and 8.82% for TGT-At. Notably, DeMol consistently achieves the highest EwT values across all categories, including In-Domain (ID), Out-of-Domain Adsorption (OOD Ads.), Out-of-Domain Catalysis (OOD Cat.), and Out-of-Domain Both

Table 2: Performance on OC20 IS2RE validation set. NN refers to "Noisy Nodes" (Godwin et al., 2021). Bold values indicate the best performance.

| Model | Energy MAE (eV) ↓ | | | | | EwT (%) ↑ | | | | |
|---|---|---|---|---|---|---|---|---|---|---|
| | ID | OOD Ads. | OOD Cat. | OOD Both | AVG. | ID | OOD Ads. | OOD Cat. | OOD Both | AVG. |
| SchNet | 0.6465 | 0.7074 | 0.6475 | 0.6626 | 0.6660 | 2.96 | 2.22 | 3.03 | 2.38 | 2.65 |
| DimeNet++ | 0.5636 | 0.7127 | 0.5612 | 0.6492 | 0.6217 | 4.25 | 2.48 | 4.40 | 2.56 | 3.42 |
| GemNet-T | 0.5561 | 0.7342 | 0.5659 | 0.6964 | 0.6382 | 4.51 | 2.24 | 4.37 | 2.38 | 3.38 |
| SphereNet | 0.5632 | 0.6682 | 0.5590 | 0.6190 | 0.6024 | 4.56 | 2.70 | 4.59 | 2.70 | 3.64 |
| Graphormer-3D | 0.4329 | 0.5850 | 0.4441 | 0.5299 | 0.4980 | - | - | - | - | - |
| GNS | 0.54 | 0.65 | 0.55 | 0.59 | 0.5825 | - | - | - | - | - |
| GNS+NN | 0.47 | 0.51 | 0.48 | 0.46 | 0.4800 | - | - | - | - | - |
| DRFormer | 0.4222 | 0.5420 | 0.4231 | 0.4754 | 0.4657 | 7.23 | 3.77 | 7.13 | 4.10 | 5.56 |
| EquiFormer+NN | 0.4156 | 0.4976 | 0.4165 | 0.4344 | 0.4410 | 7.47 | 4.64 | 7.19 | 4.84 | 6.04 |
| DRFormer | 0.4187 | 0.4863 | 0.4321 | 0.4332 | 0.4425 | 8.39 | 5.42 | 8.12 | 5.44 | 6.84 |
| Unimol+ | 0.3795 | 0.4526 | 0.4011 | 0.4021 | 0.4088 | 11.15 | 6.71 | 9.90 | 6.68 | 8.61 |
| TGT-At | 0.3813 | 0.4454 | 0.3917 | 0.3936 | 0.4030 | 11.15 | 6.87 | 10.47 | 6.80 | 8.82 |
| DeMol | **0.3663** | **0.4302** | **0.3746** | **0.3804** | **0.3879** | **12.04** | **6.98** | **10.86** | **7.01** | **9.23** |

(OOD Both). This consistent superiority highlights DeMol's robustness in handling both in-domain and diverse out-of-domain scenarios.

**QM9.** We use the QM9 dataset (Ramakrishnan et al., 2014) to evaluate our model on molecular tasks in the 3D format. QM9 is a quantum chemistry benchmark consisting of 134k stable small organic molecules. These molecules correspond to the subset of all 133,885 species out of the GDB-17 chemical universe of 166 billion organic molecules. Each molecule is associated with 12 prediction targets covering its energetic, electronic, and thermodynamic properties. The 3D geometric structure of the molecule is used as the model's input. Following (Thölke & De Fabritiis, 2022), we randomly choose 10,000 and 10,831 molecules for validation and test evaluation, respectively. The remaining molecules are used to fine-tune our DeMol model. The details of baselines and experiment settings are presented in Appendix D.4.

Table 3: Results on QM9. The evaluation metric is the Mean Absolute Error (MAE). We report the official results of baselines from Appendix D.4. Bold values indicate the best performance.

| Model | $\mu$ | $\alpha$ | $\epsilon_{HOMO}$ | $\epsilon_{LUMO}$ | $\Delta\epsilon$ | $R^2$ | ZPVE | $U_0$ | U | H | G | $C_v$ | Avg. Rank # |
|---|---|---|---|---|---|---|---|---|---|---|---|---|---|
| EdgePred | 0.039 | 0.086 | 37.4 | 31.9 | 58.2 | 0.112 | 1.81 | 14.7 | 14.2 | 14.8 | 14.5 | 0.038 | 18.92 |
| AttrMask | 0.020 | 0.072 | 31.3 | 37.8 | 50.0 | 0.423 | 1.90 | 10.7 | 10.8 | 11.4 | 11.2 | 0.062 | 15.25 |
| InfoGraph | 0.041 | 0.099 | 48.1 | 38.1 | 72.2 | 0.114 | 1.69 | 16.4 | 14.9 | 14.5 | 16.5 | 0.030 | 19.08 |
| GraphCL | 0.027 | 0.066 | 26.8 | 22.9 | 45.5 | 0.095 | 1.42 | 9.6 | 9.7 | 9.6 | 10.2 | 0.028 | 10.0 |
| GPT-GNN | 0.039 | 0.103 | 35.7 | 28.8 | 54.1 | 0.158 | 1.75 | 12.0 | 24.8 | 14.8 | 12.2 | 0.032 | 18.17 |
| GraphMVP | 0.031 | 0.070 | 28.5 | 26.3 | 46.9 | 0.082 | 1.63 | 10.2 | 10.3 | 10.4 | 11.2 | 0.033 | 12.00 |
| GEM | 0.034 | 0.081 | 33.8 | 27.7 | 52.1 | 0.089 | 1.73 | 13.4 | 12.6 | 13.3 | 13.2 | 0.035 | 16.00 |
| 3D Infomax | 0.034 | 0.075 | 29.8 | 25.7 | 48.8 | 0.122 | 1.67 | 12.7 | 12.5 | 12.4 | 13.0 | 0.033 | 14.92 |
| PosPred | 0.024 | 0.067 | 25.1 | 20.9 | 40.6 | 0.115 | 1.46 | 10.2 | 10.3 | 10.2 | 10.9 | 0.035 | 11.08 |
| 3D-MGP | 0.020 | 0.057 | 21.3 | 18.2 | 37.1 | 0.092 | 1.38 | 8.6 | 8.6 | 8.7 | 9.3 | 0.026 | 6.83 |
| SchNet | 0.033 | 0.235 | 41 | 34 | 63 | 0.073 | 1.7 | 14 | 19 | 14 | 14 | 0.033 | 17.33 |
| PhysNet | 0.0529 | 0.0615 | 32.9 | 24.7 | 42.5 | 0.765 | 1.39 | 8.15 | 8.34 | 8.42 | 9.4 | 0.028 | 11.67 |
| Cormorant | 0.038 | 0.085 | 34 | 38 | 61 | 0.961 | 2.027 | 22 | 21 | 21 | 20 | 0.026 | 20.08 |
| DimeNet++ | 0.0297 | 0.0435 | 24.6 | 19.5 | 32.6 | 0.331 | 1.21 | 6.32 | 6.28 | 6.53 | 7.56 | 0.023 | 6.33 |
| PaiNN | 0.012 | 0.045 | 27.6 | 20.4 | 45.7 | 0.066 | 1.28 | 5.85 | 5.83 | 5.98 | 7.35 | 0.024 | 4.67 |
| ALIGNN | 0.0146 | 0.0561 | 21.4 | 19.5 | 38.1 | 0.543 | 3.1 | 15.3 | 14.4 | 14.7 | 14.4 | 0.033 | 13.83 |
| LieTF | 0.041 | 0.082 | 33 | 27 | 51 | 0.448 | 2.10 | 17 | 16 | 17 | 19 | 0.035 | 19.58 |
| TorchMD-Net | **0.011** | 0.059 | 20.3 | 17.5 | 36.1 | **0.033** | 1.84 | 6.15 | 6.38 | 6.16 | 7.62 | 0.026 | 5.25 |
| EGNN | 0.029 | 0.071 | 29 | 25 | 48 | 0.106 | 1.55 | 11 | 12 | 12 | 12 | 0.031 | 13.00 |
| NoisyNode | 0.025 | 0.052 | 20.4 | 18.6 | 28.6 | 0.70 | **1.16** | 7.30 | 7.57 | 7.43 | 8.30 | 0.025 | 6.75 |
| Transformer-M | 0.037 | **0.041** | 17.5 | 16.2 | 27.4 | 0.075 | 1.18 | 9.37 | 9.41 | 9.39 | 9.63 | 0.022 | 6.00 |
| ESA | 0.024 | **0.041** | 17.4 | 16.0 | 27.1 | 0.101 | 1.20 | 6.42 | 6.62 | 6.45 | 8.39 | 0.024 | 4.17 |
| DeMol | 0.024 | 0.042 | **16.4** | **15.7** | **26.8** | 0.101 | **1.16** | 6.24 | 6.1 | 6.35 | **6.29** | **0.021** | **2.67** |

Evaluation results are presented in Table 3, demonstrating that our DeMol achieves competitive performance against strong baselines. In particular, DeMol achieves state-of-the-art performance on HOMO, LUMO, HOMO-LUMO gap, ZPVE, G, and $C_v$ predictions. DeMol also demonstrates comparable performance across other metrics. Our insight is that this result stems from the nature of the QM9 dataset itself, relative to DeMol's specific strengths. The QM9 dataset consists of 134k very small, simple organic molecules, limited to 9 heavy atoms (C, O, N, F). DeMol is explicitly designed to capture complex, non-local bond-level phenomena like resonance and nuanced 3D bond-bond interactions that determine properties like stereoselectivity. These complex phenomena are far less prevalent or impactful in the small, structurally simple molecules of QM9 compared to the larger, more diverse molecules in PCQM4Mv2 or the complex multi-component systems in OC20. These findings suggest that DeMol successfully captures essential quantum mechanical information through its modelling of atomic and bond interactions, leading to accurate predictions across diverse

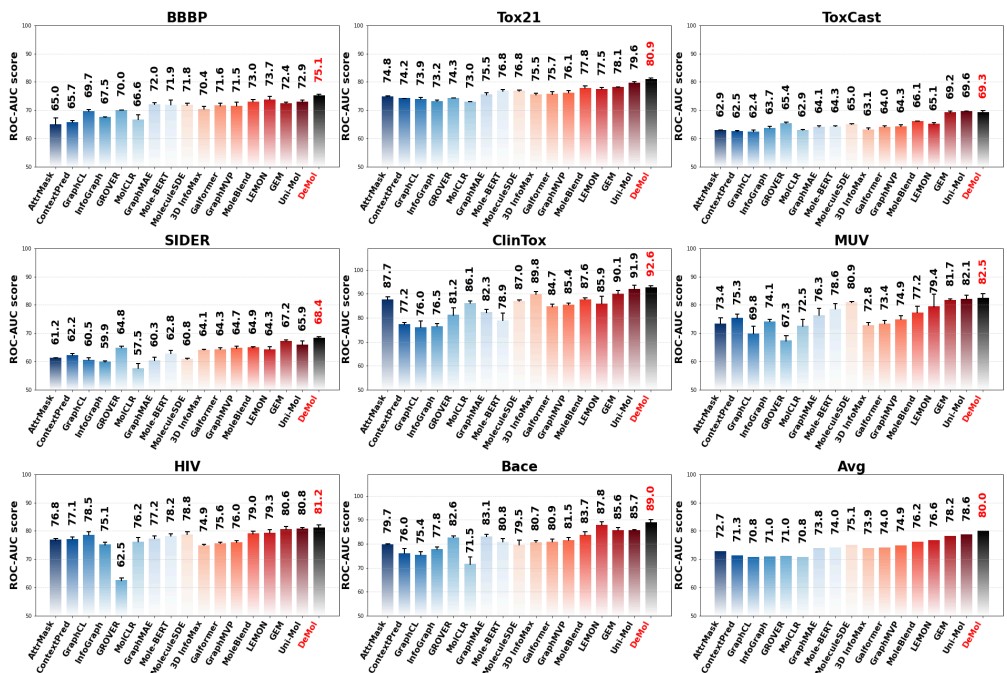

Figure 4: Results on molecular property classification tasks. The table version is Appendix Table 6.

molecular properties. It highlights the effectiveness of DeMol's architecture in learning transferable representations for multiple quantum chemistry tasks.

**MoleculeNet.** After the model is pre-trained, we evaluate our DeMol on the MoleculeNet benchmark (Wu et al., 2018), which involves eight widely-used binary classification datasets. These datasets span diverse domains, including quantum chemistry, physical chemistry, biophysics, and physiology, providing a comprehensive resource for cheminformatics modelling. Following established practice, we employ *scaffold splitting* (Ramsundar et al., 2019) to split the molecules according to their structures, ensuring the evaluation that better reflects real-world use cases. The details of baselines and experiment settings are presented in Appendix D.5.

Figure 4 and Appendix Table 6 compares the performance of molecular representation learning methods across eight MoleculeNet benchmark datasets, evaluated using ROC-AUC scores under scaffold splitting with 10 random seeds. Our DeMol achieves state-of-the-art performance with an average ROC-AUC score of 79.96, surpassing all existing baselines. The model demonstrates superior performance on 7 out of 8 benchmark datasets (BBBP, Tox21, SIDER, ClinTox, MUV, HIV, and BACE). Meanwhile, DeMol maintains competitive performance on the ToxCast datasets compared to GEM (Fang et al., 2022) and Uni-Mol (Zhou et al., 2023). Specifically, compared to Galformer (Bai et al., 2023), LEMON (Chen et al., 2025), and GEM (Fang et al., 2022), which employ bond-centric graphs as well, DeMol achieves superior performance across all molecular property prediction benchmarks. These results demonstrate DeMol's superiority and robustness across diverse chemical and biological properties. Moreover, its advantages are gained not only from its bond-centric graph representation but also from its explicit modelling of bond-bond and atom-bond interactions, which collectively enhance molecular property prediction accuracy.

## 4.1 ABLATION STUDY

We conduct a systematic ablation study to evaluate the key components of DeMol using the PCQM4Mv2 dataset, with MAE (meV) as the evaluation metric (Table 4). For a fair comparison, all hyperparameters are kept the same as the setting in the Appendix D.2. The results demonstrate that: (1) Using only the bond-centric graph achieves an MAE of 89.9 meV, while using only the atom-centric graph reduces to 77.2 meV. (2) Combining atom-centric graph and torsional angle position encoding further improves to 75.5 meV, whereas incorporating a preliminary combination of the atom-centric and bond-centric graphs significantly reduces the MAE to 65.4 meV. This highlights the complementary nature of the two graph representations.

Table 4: Ablation study on the impact of every component. Experiments are conducted on the PCQM4Mv2 dataset.

| Atom-centric Graph | Bond-centric Graph | Coval. Radii Pred. | Tors. Angle Posit. | Struc. aware Mask | Bond-bond Att. | Bond-atom Att. | Val. MAE($\downarrow$) (meV) |
|---|---|---|---|---|---|---|---|
| - | ✓ | - | ✓ | - | ✓ | - | 89.9 |
| ✓ | - | - | - | - | - | - | 77.2 |
| ✓ | - | ✓ | - | ✓ | - | - | 76.4 |
| ✓ | - | - | ✓ | - | - | - | 75.5 |
| ✓ | ✓ | ✓ | ✓ | - | ✓ | - | 65.4 |
| ✓ | ✓ | ✓ | ✓ | ✓ | ✓ | - | 64.8 |
| ✓ | ✓ | - | ✓ | ✓ | ✓ | ✓ | 61.1 |
| ✓ | ✓ | ✓ | ✓ | - | ✓ | ✓ | 61.7 |
| ✓ | ✓ | ✓ | ✓ | ✓ | ✓ | ✓ | 60.3 |

(3) Incorporating covalent radii prediction, torsional angle encoding and structure-aware masks enables explicit modelling of angular relationships and prunes non-physical interactions, leading to enhanced geometric consistency. (4) The bond-bond and bond-atom attention mechanisms provide additional gains by capturing explicit chemical bond interactions and cross-level atom-bond relationships. These findings collectively validate that each component contributes uniquely to the model's performance. The dual-graph architecture provides fundamental atom/bond representations, while geometric constraints and attention mechanisms refine these through physical plausibility enforcement and interaction modelling.

## 5 CONCLUSION

In this paper, we propose DeMol, a dual-graph enhanced multi-scale interaction framework for molecular representation learning, to address the critical gap in explicitly modelling chemical bonds and their interactions. Extensive experiments demonstrate that DeMol achieves state-of-the-art performance on diverse molecular property prediction tasks across PCQM4Mv2, OC20 IS2RE, QM9, and MoleculeNet datasets.

## ACKNOWLEDGMENTS

The research described in this paper has been partially supported by the General Research Funds from the Hong Kong Research Grants Council (project no. PolyU 15200023, 15206024, and 15224524), internal research funds from Hong Kong Polytechnic University (project no. P0042693, P0048625, and P0051361,P0059586), and Sheertek International (HK) Limited. This work was supported by computational resources provided by The Centre for Large AI Models (CLAIM) of The Hong Kong Polytechnic University.

## ETHICS STATEMENT

This work adheres to the ICLR Code of Ethics. In this study, no human subjects or animal experimentation were involved. All datasets used, including PCQM4Mv2, OC2020 IS2RE, QM9 and MoleculeNet, were sourced in compliance with relevant usage guidelines, ensuring no violation of privacy. We have taken care to avoid any biases or discriminatory outcomes in our research process. No personally identifiable information was used, and no experiments were conducted that could raise privacy or security concerns. We are committed to maintaining transparency and integrity throughout the research process.

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

**Table of Appendix:**

# A  JUSTIFICATION

## A.1  JUSTIFICATION OF PROPOSITION 1

**Proposition 1 (Information Gain from Edge Adjacency Patterns)**   *For non-trivial graphs $\mathcal{G}$, the entropy of $\mathcal{L}(\mathcal{G})$ satisfies:*

$$\mathbf{H}(\mathcal{L}(\mathcal{G})) = \mathbf{H}(\mathcal{G}) + \underbrace{\mathbf{H}(\mathcal{E}'|\mathcal{E})}_{\Delta I_{\text{edge-structure}}} \quad \text{with} \quad \Delta I_{\text{edge-structure}} > 0.$$

**Justification:**   In information theory, the information content of a graph $\mathcal{G}$ can be quantified via its structural entropy. For the atom-centric graph $\mathcal{G} = (\mathcal{V}, \mathcal{E})$, its entropy is decomposed as:

$$\mathbf{H}(\mathcal{G}) = \mathbf{H}(\mathcal{V}) + \mathbf{H}(\mathcal{E}|\mathcal{V}), \tag{11}$$

where $\mathbf{H}(\mathcal{V})$ is the entropy of the node distribution (e.g., entropy of degree distribution). $\mathbf{H}(\mathcal{E}|\mathcal{V})$ is the conditional entropy of edge connections given the vertex distribution, reflecting the randomness of edge placements. The bond-centric graph $\mathcal{L}(\mathcal{G})$ elevated edges $\mathcal{E}$ of $\mathcal{G}$ to nodes and encodes adjacency relations between edges. Its entropy is:

$$\mathbf{H}(\mathcal{L}(\mathcal{G})) = \mathbf{H}(\mathcal{E}) + \mathbf{H}(\mathcal{E}'|\mathcal{E}), \tag{12}$$

where $\mathbf{H}(\mathcal{E})$ is the entropy of the original edge distribution (e.g., diversity of edge weights), $\mathbf{H}(\mathcal{E}'|\mathcal{E})$ is the conditional entropy of edge adjacency patterns in $\mathcal{L}(\mathcal{G})$, reflecting the complexity of edge-to-edge connections.

Mutual information $I(\mathcal{G}; \mathcal{L}(\mathcal{G}))$, which measures shared information between $\mathcal{G}$ and $\mathcal{L}(\mathcal{G})$, is defined as:

$$I(\mathcal{G}; \mathcal{L}(\mathcal{G})) = \mathbf{H}(\mathcal{G}) + \mathbf{H}(\mathcal{L}(\mathcal{G})) - \mathbf{H}(\mathcal{G}; \mathcal{L}(\mathcal{G})). \tag{13}$$

Since $\mathcal{L}(\mathcal{G})$ is fully constructed from $\mathcal{G}$, $\mathbf{H}(\mathcal{G}; \mathcal{L}(\mathcal{G})) = \mathbf{H}(\mathcal{L}(\mathcal{G}))$, leading to:

$$I(\mathcal{G}; \mathcal{L}(\mathcal{G})) = \mathbf{H}(\mathcal{L}(\mathcal{G})). \tag{14}$$

This indicates that the entropy of $\mathcal{L}(\mathcal{G})$ originates entirely from $\mathcal{G}$. However, the unique information in $\mathcal{L}(\mathcal{G})$ lies in its conditional entropy $\mathbf{H}(\mathcal{E}'|\mathcal{E})$, which captures edge adjacency patterns not explicitly encoded in $\mathcal{G}$. Thus, for non-trivial graphs $\mathcal{G}$, the entropy of $\mathcal{L}(\mathcal{G})$ satisfies:

$$\mathbf{H}(\mathcal{L}(\mathcal{G})) = \mathbf{H}(\mathcal{G}) + \underbrace{\mathbf{H}(\mathcal{E}'|\mathcal{E})}_{\Delta I_{\text{edge-structure}}} \quad \text{with} \quad \Delta I_{\text{edge-structure}} > 0.$$

$\mathbf{H}(\mathcal{E}'|\mathcal{E})$ quantifies the randomness of edge adjacency patterns in $\mathcal{G}$, introducing higher-order structural information absent in $\mathcal{G}$. This guarantees $\Delta I_{\text{edge-structure}} > 0$.

## A.2 JUSTIFICATION OF PROPOSITION 2

**Proposition 2 (Mutual Information Decomposition)** *For any molecular graph $\mathcal{G}$ and its dual-graph representation $(\mathbf{H}^{(a)}, \mathbf{H}^{(b)})$, the mutual information satisfies:*

$$I(\mathcal{G}, \mathcal{L}(\mathcal{G}); \mathbf{H}^{(a)}, \mathbf{H}^{(b)}) = \underbrace{I(\mathcal{G}; \mathbf{H}^{(a)})}_{\text{Atomic Information}} + \underbrace{I(\mathcal{L}(\mathcal{G}); \mathbf{H}^{(b)}|\mathbf{H}^{(a)})}_{\text{Bond-centric graph Information}} + \underbrace{I(\mathcal{G}; \mathbf{H}^{(b)}|\mathcal{L}(\mathcal{G}), \mathbf{H}^{(a)})}_{\text{Residual Atomic Dependency}}.$$

**Justification:**

**Lemma 1 (Chain rule of mutual information) (Cover et al., 1991)**
$$I(X, Z; Y) = I(X; Y) + I(Z; Y|X)$$

$$
\begin{aligned}
I(X, Z; Y) &= E_{p(x,z,y)}\Big[\log \frac{p(z,y,x)}{p(y)p(z,x)}\Big] \\
&= E_{p(x,z,y)}\Big[\log \frac{p(x,y)p(z,y,x)}{p(y)p(z,x)p(y,x)}\Big] \\
&= E_{p(x,z,y)}\Big[\log \frac{p(x,y)}{p(x)p(y)} \frac{p(z,y|x)}{p(z|x)p(y|x)}\Big] \\
&= E_{p(x,y)}\Big[\log \frac{p(x,y)}{p(x)p(y)}\Big] + E_{p(x,z,y)}\Big[\log \frac{p(z,y|x)}{p(z|x)p(y|x)}\Big] \\
&= I(X; Y) + I(Z; Y|X)
\end{aligned}
\tag{15}
$$

We start with the total mutual information $I(\mathcal{G}, \mathcal{L}(\mathcal{G}); \mathbf{H}^{(a)}, \mathbf{H}^{(b)})$ and apply the chain rule for mutual information:

$$I(\mathcal{G}, \mathcal{L}(\mathcal{G}); \mathbf{H}^{(a)}, \mathbf{H}^{(b)}) = I(\mathcal{G}, \mathcal{L}(\mathcal{G}); \mathbf{H}^{(a)}) + I(\mathcal{G}, \mathcal{L}(\mathcal{G}); \mathbf{H}^{(b)}|\mathbf{H}^{(a)}).$$

Since $\mathcal{L}(\mathcal{G})$ is a deterministic function of $\mathcal{G}$, knowing $\mathcal{G}$ provides no additional information about $\mathcal{L}(\mathcal{G})$. Therefore, $I(\mathcal{L}(\mathcal{G}); \mathbf{H}^{(a)}|\mathcal{G}) = 0$. This simplifies the first term:

$$I(\mathcal{G}, \mathcal{L}(\mathcal{G}); \mathbf{H}^{(a)}) = I(\mathcal{G}; \mathbf{H}^{(a)}) + I(\mathcal{L}(\mathcal{G}); \mathbf{H}^{(a)}|\mathcal{G}) = I(\mathcal{G}; \mathbf{H}^{(a)}).$$

The second term, $I(\mathcal{G}, \mathcal{L}(\mathcal{G}); \mathbf{H}^{(b)}|\mathbf{H}^{(a)})$, can be expanded as:

$$I(\mathcal{G}, \mathcal{L}(\mathcal{G}); \mathbf{H}^{(b)}|\mathbf{H}^{(a)}) = I(\mathcal{L}(\mathcal{G}); \mathbf{H}^{(b)}|\mathbf{H}^{(a)}) + I(\mathcal{G}; \mathbf{H}^{(b)}|\mathcal{L}(\mathcal{G}), \mathbf{H}^{(a)}).$$

Combining these gives the full decomposition:

$$I(\mathcal{G}, \mathcal{L}(\mathcal{G}); \mathbf{H}^{(a)}, \mathbf{H}^{(b)}) = I(\mathcal{G}; \mathbf{H}^{(a)}) + I(\mathcal{L}(\mathcal{G}); \mathbf{H}^{(b)}|\mathbf{H}^{(a)}) + I(\mathcal{G}; \mathbf{H}^{(b)}|\mathcal{L}(\mathcal{G}), \mathbf{H}^{(a)}).$$

This decomposition demonstrates that dual-graph learning retains strictly more information than single-graph approaches when $I(\mathcal{L}(\mathcal{G}); \mathbf{H}^{(b)}|\mathbf{H}^{(a)}) > 0$ and $I(\mathcal{G}; \mathbf{H}^{(b)}|\mathcal{L}(\mathcal{G}), \mathbf{H}^{(a)}) > 0$.

## A.3 JUSTIFICATION OF PROPOSITION 3

**Proposition 3 (Geometric Information Gain)** *Let $\theta_{ijk}$ denote bond angles and $\varphi_{ijkl}$ denote dihedral angles. If $\mathcal{L}(\mathcal{G})$ encodes angular relationships through $\mathcal{E}'$, the bond modelling gain satisfies:*

$$\Delta I = I(\mathcal{L}(\mathcal{G}); \mathbf{H}^{(b)}|\mathcal{G}; \mathbf{H}^{(a)}) \propto \mathbb{E}_{i,j,k,l}\left[\log \frac{p(\theta_{ijk}, \varphi_{ijkl}|\mathbf{h_i}^{(a)}, \mathbf{h_j}^{(a)}, \mathbf{h_k}^{(a)}, \mathbf{h_l}^{(a)})}{p(\theta_{ijk}, \varphi_{ijkl})}\right]. \tag{16}$$

**Justification:** This proposition provides a rationale for why incorporating a bond-centric graph $\mathcal{L}(\mathcal{G})$ enhances the geometric richness of the learned molecular representations. The core of this justification lies in connecting the structural properties of $\mathcal{L}(\mathcal{G})$ to its capacity for encoding higher-order geometric information, which is then quantified using an information-theoretic framework.

1. **Defining the Geometric Information Gain** ($\Delta I$): The term of interest, $\Delta I = I(\mathcal{L}(\mathcal{G}); \mathbf{H}^{(b)}|\mathcal{G}; \mathbf{H}^{(a)})$, is a conditional mutual information. It quantifies the amount of additional information that the bond-centric graph $\mathcal{L}(\mathcal{G})$ provides about the bond representations $\mathbf{H}^{(b)}$, given that the atom-centric graph $\mathcal{G}$ and its corresponding atom representations $\mathbf{H}^{(a)}$ are already known. In the context of our framework, this term represents the marginal information gain specifically attributable to the bond-centric perspective, which we posit is predominantly geometric in nature.

2. **Structural Suitability of $\mathcal{L}(\mathcal{G})$ for Geometric Encoding**: The atom-centric graph $\mathcal{G}$ explicitly models atomic connectivity (0-hop and 1-hop relationships). In contrast, the bond-centric graph $\mathcal{L}(\mathcal{G})$ elevates bonds to nodes, thereby making relationships between bonds explicit.

   - A bond angle ($\theta_{ijk}$) is fundamentally a property defined by two adjacent bonds sharing a common atom. In $\mathcal{L}(\mathcal{G})$, this corresponds to a direct edge between two nodes.
   - A dihedral angle ($\varphi_{ijkl}$) describes the relationship across three consecutive bonds. In $\mathcal{L}(\mathcal{G})$, this corresponds to a 2-hop path.

   Therefore, the topology of $\mathcal{L}(\mathcal{G})$ is inherently better suited to explicitly represent these higher-order geometric features than $\mathcal{G}$, where such information is only implicit and must be inferred.

3. **Quantifying Information Gain via Predictive Power**: A central tenet of representation learning is that a high-quality embedding should capture the salient properties of the input data. For molecules, 3D geometry is a fundamental property. Consequently, the quality of the learned representations ($\mathbf{H}^{(a)}, \mathbf{H}^{(b)}$) can be measured by their ability to reconstruct or predict these geometric properties. The expression on the right-hand side of the proposition is the Kullback-Leibler (KL) divergence, $D_{KL}(p(\theta, \varphi|\mathbf{H})||p(\theta, \varphi))$, averaged over the data distribution. This KL divergence measures the reduction in uncertainty about the geometric variables ($\theta, \varphi$) after observing the learned representation $\mathbf{H}$.

The structural nature of $\mathcal{L}(\mathcal{G})$ enables the model to learn a bond representation $\mathbf{H}^{(b)}$ that is more richly infused with geometric information. The degree to which these representations have captured geometric information is directly measured by their power to predict the true geometric parameters ($\theta_{ijk}, \varphi_{ijkl}$). A more informative representation will lead to a posterior distribution $p(\theta, \varphi|\mathbf{H})$ that is sharply peaked around the true values, resulting in a larger KL divergence from the uninformative prior $p(\theta, \varphi)$. Thus, the geometric information gain, $\Delta I$, derived from the bond-centric channel is directly proportional to the model's enhanced ability to predict these geometric quantities. This justifies our architectural choice to integrate geometric features like torsional angles specifically within the bond-centric channel of our framework.

### A.4 JUSTIFICATION OF PROPOSITION 4

**Proposition 4 (Dual-Graph Information Bottleneck)** *For molecular property prediction $\mathbf{Y}$, the optimal encoder $\mathbf{\Phi}$ minimizes:*

$$\min_{\mathbf{\Phi}} \left[ I(\mathcal{G}, \mathcal{L}(\mathcal{G}); \mathbf{H}^{(a)}, \mathbf{H}^{(b)}) - \beta I(\mathbf{H}^{(a)}, \mathbf{H}^{(b)}; \mathbf{Y}) \right]. \tag{17}$$

**Justification:** This proposition frames the task of learning molecular representations within the **Information Bottleneck (IB) principle**. The objective is to demonstrate that a dual-graph input provides a more advantageous starting point for solving the IB optimisation problem compared to a single, atom-centric graph.

1. **The Information Bottleneck Principle**: The IB principle posits that an optimal representation (or encoding) of an input variable $\mathbf{X}$ for predicting a target variable $\mathbf{Y}$ should satisfy two competing objectives:

- **Compression**: It should compress the input $\mathbf{X}$ as much as possible, discarding non-essential information. This is achieved by minimizing the mutual information $I(\mathbf{X}; \mathbf{Z})$ between the input $\mathbf{X}$ and the representation $\mathbf{Z}$.
- **Prediction**: It should retain as much information as possible about the target $\mathbf{Y}$. This is achieved by maximizing the mutual information $I(\mathbf{Z}; \mathbf{Y})$.

The Lagrangian multiplier $\beta$ controls the trade-off between these two objectives.

2. **Mapping the IB Principle to the DeMol Framework**: In our context, the variables are mapped as follows:
   - The input $\mathbf{X}$ corresponds to the complete dual-graph description of the molecule: $(\mathcal{G}, \mathcal{L}(\mathcal{G}))$.
   - The learned representation $\mathbf{Z}$ corresponds to the set of atom and bond embeddings: $(\mathbf{H}^{(a)}, \mathbf{H}^{(b)})$.
   - The target variable $\mathbf{Y}$ is the molecular property to be predicted.

   Thus, the objective function in Proposition 4 is a direct application of the IB principle to our framework, where the encoder $\Phi$ learns the mapping from the dual-graph input to the latent embeddings.

3. **The Advantage of a Dual-Graph Input**: The central argument for the superiority of the dual-graph approach within the IB framework is that it provides a richer, more structured input to the encoder.
   - **Richer Information Content**: As established in **Proposition 1**, the dual-graph input $(\mathcal{G}, \mathcal{L}(\mathcal{G}))$ contains strictly more information than the atom-centric graph $\mathcal{G}$ alone, particularly regarding explicit geometric and relational bond information.
   - **Improved Compression-Prediction Trade-off**: By starting with a more informative and structured input, the encoder $\Phi$ s better positioned to find a more optimal solution to the IB trade-off. The model can more effectively disentangle which aspects of the molecular structure are predictive of the target property $\mathbf{Y}$ (e.g., a specific torsional angle critical for bioactivity) from those that are merely structural artifacts. An encoder operating only on $\mathcal{G}$ might be forced to retain more ambiguous structural information because the critical geometric cues are only implicitly represented. The dual-graph input allows the encoder to be more selective, potentially achieving a representation that is simultaneously more compressive (by discarding redundant atomic information) and more predictive (by focusing on the explicit bond-level features that matter). This leads to a better trade-off, or a tighter bound, in the optimization process.

4. **Connection to Architectural Choices**: This theoretical motivation directly informs our model's design. The IB principle rationalizes not only the end-to-end learning objective but also the necessity of regularization to achieve effective compression. The **Structure-aware Mask** and the **Bond Prediction based on Covalent Radii** module can be interpreted as forms of inductive bias that aid the compression objective. By enforcing chemical and physical plausibility, these components guide the encoder to ignore vast regions of the potential representation space that correspond to unrealistic molecular conformations, thereby helping it find a more compact and meaningful representation.

In summary, Proposition 4 justifies our dual-graph framework by positioning it within the established Information Bottleneck principle. The richer input provided by the dual-graph representation allows for a more efficient and effective optimization of the compression-prediction trade-off, ultimately leading to learned embeddings that are both concise and highly predictive of molecular properties.

## B    BOND PREDICTION BASED ON COVALENT RADII ALGORITHM

## C    TIME COMPLEXITY ANALYSIS

In this section, we analyse the time complexity of the DeMol framework. This analysis helps to understand the computational efficiency of our dual-graph architecture and provides insights into how the proposed design choices affect scalability.

The DeMol framework operates on a dual-graph representation, consisting of:

---

**Algorithm 1:** Bond Prediction based on Covalent Radii.

---

**Input:** List of $N$ atoms of molecule $\mathcal{M}$: $A = \{a_1, a_2, ..., a_N\}$; Each atomic type (e.g., 'H', 'C', 'O'): $T$; 3D coordinates: $\vec{p} = (x, y, z)$; Covalent radii dictionary: $R_{cov}$; Bonding threshold factor:$\alpha = 1.15$.
**Output:** Predited Bonds $B$.

---

1   $B \leftarrow \emptyset$;      $\triangleright$ Initialize an empty set for bonds
2   **for** $i \leftarrow 1$ **to** $N - 1$ **do**
3     $T_i \leftarrow \text{type}(a_i)$;      $\triangleright$ Get atom type
4     $r_i \leftarrow R_{cov}[T_i]$;      $\triangleright$ Look up covalent radii
5     $\vec{p}_i \leftarrow (x_i, y_i, z_i)$;      $\triangleright$ Get coordinate
6     **for** $j \leftarrow i + 1$ **to** $N$ **do**
7       $T_j \leftarrow \text{type}(a_j)$;      $\triangleright$ Get atom type
8       $r_j \leftarrow R_{cov}[T_j]$;      $\triangleright$ Look up covalent radii
9       $\vec{p}_j \leftarrow (x_j, y_j, z_j)$;      $\triangleright$ Get coordinate
10      $R_{sum} \leftarrow r_i + r_j$;      $\triangleright$ Calculate the sum of covalent radii
11      $D_{threshold} \leftarrow \alpha \times R_{sum}$;      $\triangleright$ Calculate the bonding distance threshold
12      $D_{ij} \leftarrow \sqrt{(x_i - x_j)^2 + (y_i - y_j)^2 + (z_i - z_j)^2}$;      $\triangleright$ Calculate the Euclidean distance
13      **if** $D_{ij} \leq D_{threshold}$ **then**
14        $B \leftarrow B \cup \{(i, j)\}$;      $\triangleright$ Add the bond pair to the set
15      **end**
16     **end**
17 **end**
18 **return** $B$;      $\triangleright$ Return the set of predicted bonds

---

1. An atom-centric graph $\mathcal{G} = (\mathcal{V}, \mathcal{E})$, where nodes represent atoms and edges represent chemical bonds.

2. A bond-centric graph $\mathcal{L}(\mathcal{G})$, derived from $\mathcal{G}$, where nodes represent bonds and edges encode spatial or topological relationships between bonds.

Let $|\mathcal{V}| = N$ denote the number of atoms in a molecule and $|\mathcal{E}| = M$ the number of bonds. The feature dimensions for atoms and bonds are denoted as $d$ and $d_b$, respectively. We assume the use of multi-head attention with $h$ heads and $L$ layers for both the atom-centric and bond-centric channels.

**Atom-Centric Graph.** Each layer of the atom-centric graph processing involves a graph attention mechanism. For a sparse molecular graph, the number of edges $M \sim O(N)$. Hence, the time complexity per layer is $O(h \cdot M \cdot d)$. For $L$ layers, the total complexity becomes $O(L \cdot h \cdot N \cdot d)$.

**Bond-Centric Graph.** The bond-centric graph has $M$ nodes (one per bond), and its edge set $\mathcal{E}'$ encodes adjacency between bonds. In the worst case, $|\mathcal{E}'| \sim O(|\mathcal{E}|^2)$, especially when considering all possible spatial interactions between bonds. Therefore, each layer of the bond-centric channel has a complexity of $O(h \cdot |\mathcal{E}'| \cdot d_b) = O(h \cdot |\mathcal{E}|^2 \cdot d_b) = O(h \cdot M^2 \cdot d_b)$. With $L$ layers, the total complexity becomes $O(L \cdot h \cdot M^2 \cdot d_b)$.

**Cross-level Attention.** To align representations across graphs, DeMol employs cross-level attention between atoms and bonds. This introduces additional pairwise attention computations. Assuming full interaction between atoms and bonds, the complexity is $O(h \cdot N \cdot M \cdot d)$. This term dominates the overall complexity in large molecules due to the quadratic growth in $M$ and the product $N \cdot M$. Combining the above components, the overall time complexity of DeMol is $O(Lh(Nd + M^2 d_b + NMd))$.

**With Structure-aware Mask.** To mitigate the high computational overhead, we introduce the structure-aware mask, which restricts attention computation to only physically plausible atomic and bond interactions based on chemical valency rules and covalent radii constraints.

**Updated Complexity Components.**

1. Atom-Centric Graph: With the mask applied, the number of edges remains linear in $N$, so the complexity stays at $O(LhNd)$.

2. Bond-Centric Graph: Due to the mask, the number of edges in the bond-centric graph is reduced from $O(|\mathcal{E}|^2) \rightarrow O(|\mathcal{E}|)$, yielding $O(Lh|\mathcal{E}|d_b)$.

3. Cross-Level Attention. Similarly, the number of valid atom–bond interactions is now proportional to $|\mathcal{E}|$ instead of $N \cdot |\mathcal{E}|$, leading to $O(LhKd)$, where $K \ll N \cdot |\mathcal{E}|$ denotes the number of masked interactions.

**Total Complexity.**  Under the structure-aware masking strategy, the overall time complexity of DeMol becomes $O(Lh(Nd + |\mathcal{E}|d_b)) = O(Lh(Nd + Md_b))$. In fact, most of molecule graphs satisfy the condition that the number of edges is 1.1 to 1.2 times the number of atoms, $M \approx 1.2N$. Then, we get the total complexity $O(Lh(Nd + Md_b)) = O(Lh(Nd + N^2d_b + N^2d)) = O(Lh(Nd + (d + d_b)N^2)))$.

## C.1 INFERENCE TIME ANALYSIS

We further provide additional results on the inference time on the PCQM4Mv2 dataset on an NVIDIA A6000 GPU.

Table 5: Inference Time Comparison on PCQM4Mv2 dataset.

| Methods | Inference Time |
|---|---|
| Transformer-M | $\sim$28 ms/molecule |
| GPS++ (Ensemble) | $\sim$100 ms/molecule |
| DeMol | $\sim$34 ms/molecule |

As shown in the Table 5, DeMol is approximately 20% slower than Transformer-M due to the additional bond-channel computations but provides a significant performance gain. It is vastly faster than ensemble-based approaches (like GPS++) while achieving superior single-model performance.

## D  EXPERIMENT DETAILS

### D.1 LARGE-SCALE PRE-TRAINING

**Dataset.**  Our DeMol model is pre-trained on the training set of PCQM4Mv2 from the OGB Large-Scale Challenge (Hu et al., 2021). PCQM4Mv2 is a quantum chemistry dataset originally curated under the PubChemQC project (Maho, 2015; Nakata & Shimazaki, 2017). The total number of training samples is 3.37 million. Each molecule in the training set is associated with both 2D graph structures and 3D geometric structures. The HOMO-LUMO energy gap of each molecule is provided, which is obtained by DFT-based geometry optimization (Burke, 2012). According to the OGB-LSC (Hu et al., 2021), the HOMO-LUMO energy gap is one of the most practically-relevant quantum chemical properties of molecules since it is related to reactivity, photoexcitation, and charge transport. Being the largest publicly available dataset for molecular property prediction, PCQM4Mv2 is considered to be a challenging benchmark for molecular models.

**Settings.**  Similar to Graphormer (Ying et al., 2021), Transformer-M (Luo et al., 2022) and Uni-mol (Zhou et al., 2023), DeMol comprises 12 layers with atom representation dimension of $d_a = 768$ and bond representation dimension of $d_b = 768$. The model employ 128 Gaussian kernels. We also use AdamW (Diederik, 2014) as the optimizer and set its hyperparameter $\epsilon$ to 1e-8 and $(\beta_1, \beta_2)$ to (0.9,0.999). The gradient clip norm is set to 5.0. The peak learning rate is set to 2e-4. The batch size is set to 1024. The model is trained for 1.5million steps with 150K warmup steps. We also utilized exponential moving average (EMA) with a decay rate of 0.999. The training process required around 7 days, powered by 8 NVIDIA A6000 GPUs.

**Pretraining strategies.**  As illustrated in Section 3 Figure 3, DeMol employs a Multi-Task Pretraining Strategy.

We follow Transformer-M (Luo et al., 2022) and Unimol (Zhou et al., 2023) settings. The primary goal is the supervised regression of the quantum property (HOMO-LUMO gap), and we also utilise auxiliary tasks to enforce geometric and structural understanding.

**Property Prediction Loss** ($\mathcal{L}_{prop}$). This is the primary supervised task. For the PCQM4Mv2 dataset, we aim to minimize the Mean Absolute Error (MAE) between the predicted HOMO-LUMO gap $\hat{y}$ and the ground truth $y$:

$$\mathcal{L}_{prop} = \frac{1}{|\mathcal{B}|} \sum_{i \in \mathcal{B}} |y_i - \hat{y}_i|,$$

where $\mathcal{B}$ is the batch of molecule.

**Masked Atom Prediction Loss** ($\mathcal{L}_{mask}$). To capture topological context, we randomly mask a portion of atomic node features. The masked atom prediction head predicts the atom type $t_i$ using Cross-Entropy loss:

$$\mathcal{L}_{mask} = - \sum_{j \in \mathcal{M}} \log P(t_j | \mathcal{G}_{\text{masked}}),$$

where $\mathcal{M}$ is the set of masked indices.

**Coordinate Recovery Loss** ($\mathcal{L}_{coord}$). To enforce 3D spatial awareness, we add noise to the input coordinates. The coordinates recovery head is trained to denoise and recover the ground-truth coordinates $\vec{p}_i$, typically minimised via Mean Squared Error (MSE):

$$\mathcal{L}_{coord} = \sum_{i=1}^{N} ||\vec{p}_i - \hat{\vec{p}}_i||^2$$

.

**Bond Prediction based on Covalent Radii** ($\mathcal{L}_{bond}$). As described in Algorithm 1, we determine the ground truth chemical bonds $B_{gt}$ based on covalent radii constraints ($D_{ij} < \alpha(r_i + r_j)$). The bonding prediction head learns to classify valid interactions, serving as a regularisation term to penalise geometrically inconsistent structures:

$$\mathcal{L}_{bond} = - \sum_{(i,j) \in \mathcal{E}_{all}} \left[ \mathbb{I}_{(i,j) \in B_{gt}} \log(p_{ij}) + (1 - \mathbb{I}_{(i,j) \in B_{gt}}) \log(1 - p_{ij}) \right],$$

where $p_{ij}$ is the predicted probability of a bond existing between atoms $i$ and $j$.

### D.2 PCQM4Mv2

**Baselines.** We compare our DeMol with several competitive baselines. These models fall into two categories: message passing neural network (MPNN) variants and Graph Transformers.

For MPNN variants, we include two widely used models, GCN (Jiang et al., 2019) and GIN (Xu et al., 2018), and their variants with virtual node (VN) (Gilmer et al., 2017; Jiang et al., 2019). Additianlly, we compare GINE-$_{VN}$ (Brossard et al., 2020; Gilmer et al., 2017; Luo et al., 2022) and DeeperGCN-$_{VN}$ (Li et al., 2020; Luo et al., 2022). GINE is the multi-hop version of GIN. DeeperGCN is a 12-layer GNN model with carefully designed aggregators. The results of MLP-Fingerprint (Hu et al., 2021) is also reported.

We also compare several Graph Transformer models. Graphormer (Ying et al., 2021) developed graph structural encodings and integrated them into a standard Transformer model. TokenGT (Kim et al., 2022) adopted the standard Transformer architecture without graph-specific modifications. GraphGPS (Rampášek et al., 2022) proposed a framework to integrate the positional and structural encodings, local message-passing mechanism, and global attention mechanism into the Transformer model. GPS++ (Masters et al., 2022) optimsed hybrid MPNN/Transformer for molecular property prediciton. GRPE (Park et al., 2022) proposed a graph-specific relative positional encoding and considered both node-spatial and node-edge relations. EGT (Hussain et al., 2022) exclusively used global self-attention as an aggregation mechanism rather than static localized convolutional aggregation, and utilized edge channels to capture structural information. GEM-2Liu et al. (2022a) models full-range many-body interactions in molecules, leveraging an axial attention mechanism to efficiently capture complex quantum interactions.Transformer-M (Luo et al., 2022) integrated 2D

and 3D spatial encodings as the attention bias to enhance molecule representation. Unimol+ (Lu et al., 2023) proposed a molecule conformer update mechanism to accurately predict quantum chemical properties. TGT (Hussain et al., 2024) integrated triplet interactions to improve the graph transformers.

### D.3  OPEN CATALYST 2020 IS2RE

The Open Catalyst 2020 Challenge (Chanussot et al., 2021) is aimed at predicting the adsorption energy of molecules on catalyst surfaces using machine learning. We focus on the IS2RE (Initial Structure to Relaxed Energy) task, where the model is provided with an initial DFT structure of the crystal and adsorbate, which interact with each other to reach the relaxed structure when the relaxed energy of the system is measured. It comprises approximately 460K training data points. While DFT equilibrium confirmations are provided for training, they are not permitted for use during inference.

**Settings.**   We use the default 12-layer setting for OC20 experiments. Firstly, since OC20 lacks graph information, graph-related features are excluded from the model. We adopt the solution proposed in  (Shi et al., 2022) to consider the periodic boundary condition, which pre-expands the neighbor cells and then applies a radius cutoff to reduce the number of atoms. The AdamW optimizer was employed during the training process, which lasted for 1.5 million steps, including 150K warmup steps. The optimizer was configured with a learning rate of 2e-4, a batch size of 64, $(\beta_1, \beta_2)$ values of (0.9,0.999), and a gradient clipping parameter of 5.0. The training process spanned approximately 14 days and make use of 8 NVIDIA A600 GPUs.

**Baselines.**   We compare our DeMol with several competitive baselines. The baselines are categorised into two groups: 3d-based models and graph-based models. SchNet (Schütt et al., 2017) proposed a pioneering 3D convolutional neural network that uses continuous-filter convolutions to model atomic interactions based on interatomic distances. DimeNet++ (Sriram et al., 2022) incorporates directional message passing to better capture angular dependencies between atoms. GemNet-T (Gasteiger et al., 2021) introduces a graph neural network designed to handle both geometric and electronic properties of molecules, focusing on translational equivariance. SphereNet (Liu et al., 2022b) represents molecules as spherical harmonics, enabling efficient computation of rotational equivariant features. For grpah-based models, Graphormer-3D (Shi et al., 2022) combines transformer architectures with 3D graph representations to capture long-range dependencies in molecular structures. GNS (Godwin et al., 2021) uses a generative neural simulator that learns to predict molecular dynamics by modeling interactions between atoms. DRFormer (Wang et al., 2023a) captures relational information in molecular graphs for drug discovery. EquiFormer (Liao & Smidt, 2022) proposes an equivariant transformer architecture that preserves symmetries in molecular data while incorporating neural network layers for enhanced performance. DRFormer (Wang et al., 2023a) is optimised for robustness and efficiency in molecular property prediction. UniMol+ (Lu et al., 2023) combines multiple modalities (e.g., 3D coordinates and graph representations) to improve generalization across diverse molecular datasets. TGT (Hussain et al., 2024) integrated triplet interactions to improve the graph transformers.

### D.4  QM9

We use the QM9 dataset (Ramakrishnan et al., 2014) to evaluate our model on molecular tasks in the 3D data format. QM9 is a quantum chemistry benchmark consisting of 134k stable small organic molecules. These molecules correspond to the subset of all 133,885 species out of the GDB-17 chemical universe of 166 billion organic molecules. Each molecule is associated with 12 targets covering its energetic, electronic, and thermodynamic properties. The 3D geometric structure of the molecule is used as input. Following  (Thölke & De Fabritiis, 2022), we randomly choose 10,000 and 10,831 molecules for validation and test evaluation, respectively. The remaining molecules are used to fine-tune our DeMol model. We observed that several previous works used different data splitting ratios or did not describe the evaluation details. For a fair comparison, we choose baselines that use similar splitting ratios in the original papers.

**Settings.**   We fine-tune the pre-trained DeMol on the QM9 dataset. Following Transformer-M (Luo et al., 2022), we adopt the Mean Squared Error (MSE) loss during training and use the Mean Absolute

Error (MAE) loss function during evaluation. We also adopt label standardisation for stable training. We use AdamW as the optimiser, and set the hyperparameter $\epsilon$ to 1e-8 and $(\beta_1, \beta_2)$ to (0.9,0.999). The gradient clip norm is set to 5.0. The peak learning rate is set to 7e-5. The batch size is set to 128. The dropout ratios for the input embeddings, attention matrices, and hidden representations are set to 0.0,0.1, and 0.0, respectively. The weight decay is set to 0.0. The model is fine-tuned for 600k steps with a 60K-step warmup stage. After the warmup stage, the learning rate decays linearly to zero. All model are trained on 8 NVIDIA A6000 GPUs.

**Baselines.** We comprehensively compare our DeMol with both pre-training methods and 3D molecular models. First, we follow (Luo et al., 2022) to compare several pre-training methods. AttrMask (Hu et al., 2019) proposed a strategy to pre-train GNNs via both node-level and graph-level tasks. InfoGraph (Sun et al., 2019) maximized the mutual information between graph-level representations and substructure representations as the pre-training tasks. GraphCL (You et al., 2020) instead used contrastive learning to pre-train GNNs. There are also several works that utilise 3D geometric structures during pre-training. GraphMVP (Jiang et al., 2019) maximized the mutual information between 2D and 3D representations. GEM (Fang et al., 2022) proposed a strategy to learning spatial information by utilizing both local and global 3D structure. 3D Infomax (Stärk et al., 2022) used two encoder to capture 2D and 3D structural information separately while maximizing the mutual information between 2D and 3D representations. PosPred (Jiao et al., 2023) adopted an equivariant energy-based model and developed a node-level pertaining loss for force prediction. We also follow (Thölke & De Fabritiis, 2022) to compare 3D molecular models. SchNet (Schütt et al., 2017) used continuous-filter convolution layers to model quantum interactions in molecule. Cormorant (Anderson et al., 2019) developed a GNN model equipped with activation functions being covariant to rotations. DimeNet++ (Gasteiger et al., 2020) proposed directional message passing, which uses atom-pair embeddings and utilizes directional information between atoms. PaiNN (Schütt et al., 2021) proposed the polarizable atom interaction neural network that uses an equivariant message passing mechanism. LieTF (Hutchinson et al., 2021) built upon the Transformer model consisting of attention layers that are equivariant to arbitrary Lie groups and other discrete subgroups. TorchMD-Net (Thölke & De Fabritiis, 2022) also developed a Transformer variant with layers designed by prior physical and chemical knowledge. EGNN (Satorras et al., 2021) proposed a model which does not require computationally expensive high-order representations in immediate layers to keep equivariance, and can be easily scaled to higher-dimensional spaces. NoisyNode (Godwin et al., 2021) proposed the 3D position denoising task and verified it on the Graph Network-based Simulator (GPS) model (Masters et al., 2022). Transformer-M (Luo et al., 2022) integrated 2D and 3D spatial encodings as the attention bias to enhance molecule representation. ALIGNN (Choudhary & DeCost, 2021) uses a line graph, treating it as a route for message passing. ESA (Buterez et al., 2025) introduces an end-to-end attention architecture that treats graphs as sets of edges to explicitly learn edge representations.

## D.5 MOLECULENET

MoleculeNet (Wu et al., 2018) is a popular benchmark for molecular property prediction, including datasets focusing on different molecular properties, from quantum mechanics and physical chemistry to biophysics and physiology. The details of the datasets we used are described below.

**BBBP.** The BBBP dataset is designed to predict whether a molecule can cross the blood-brain barrier (BBB), a critical factor in drug delivery and efficacy. This dataset consists of 2,039 molecules, where each molecule is labelled as either permeable or non-permeable based on experimental data. The task involves binary classification, requiring models to identify structural features that influence BBB permeability. The dataset is widely used to evaluate the ability of machine learning models to capture subtle molecular properties that affect BBB penetration, making it a benchmark for tasks related to drug discovery and pharmacokinetics.

**Tox21.** The Tox21 dataset focuses on predicting the toxicity of chemical compounds. It contains over 8,000 molecules, each associated with 12 different toxicity endpoints, such as nuclear receptor activation and stress response pathways. The dataset is derived from high-throughput screening experiments conducted by the U.S. Environmental Protection Agency (EPA) and the National Institutes of Health (NIH). Tox21 challenges models to accurately classify molecules based on their

Table 6: Results on molecular property classification tasks. We report the (mean±standard deviation) ROC-AUC score (higher is better) of 10 random seeds under scaffold splitting. The best results are highlighted in bold.

| Methods | BBBP ↑ | Tox21 ↑ | ToxCast ↑ | SIDER ↑ | ClinTox ↑ | MUV ↑ | HIV ↑ | BACE ↑ | Avg ↑ |
|---|---|---|---|---|---|---|---|---|---|
| AttrMask | 65.0±2.3 | 74.8±0.2 | 62.9±0.1 | 61.2±0.1 | 87.7±1.1 | 73.4±2.0 | 76.8±0.5 | 79.7±0.3 | 72.68 |
| ContextPred | 65.7±0.6 | 74.2±0.0 | 62.5±0.3 | 62.2±0.5 | 77.2±0.8 | 75.3±1.5 | 77.1±0.8 | 76.0±2.0 | 71.28 |
| GraphCL | 69.7±0.6 | 73.9±0.6 | 62.4±0.5 | 60.5±0.8 | 76.0±2.6 | 69.8±2.6 | 78.5±1.2 | 75.4±1.4 | 70.78 |
| InfoGraph | 67.5±0.1 | 73.2±0.4 | 63.7±0.5 | 59.9±0.3 | 76.5±1.0 | 74.1±0.7 | 75.1±0.9 | 77.8±0.8 | 70.98 |
| GROVER | 70.0±0.10 | 74.3±0.1 | 65.4±0.4 | 64.8±0.6 | 81.2±3.0 | 67.3±1.8 | 62.5±0.9 | 82.6±0.7 | 71.01 |
| MolCLR | 66.6±1.8 | 73.0±0.1 | 62.9±0.3 | 57.5±1.7 | 86.1±0.9 | 72.5±2.3 | 76.2±1.5 | 71.5±3.1 | 70.79 |
| GraphMAE | 72.0±0.6 | 75.5±0.6 | 64.1±0.3 | 60.3±1.1 | 82.3±1.2 | 76.3±2.4 | 77.2±1.0 | 83.1±0.9 | 73.85 |
| Mole-BERT | 71.9±1.6 | 76.8±0.5 | 64.3±0.2 | 62.8±1.1 | 78.9±3.0 | 78.6±1.8 | 78.2±0.8 | 80.8±1.4 | 74.04 |
| MoleculeSDE | 71.8±0.7 | 76.8±0.3 | 65.0±0.2 | 60.8±0.3 | 87.0±0.5 | 80.9±0.3 | 78.8±0.9 | 79.5±2.1 | 75.07 |
| 3D InfoMax | 70.4±1.0 | 75.5±0.5 | 63.1±0.7 | 64.1±0.1 | 89.8±1.2 | 72.8±1.0 | 74.9±0.3 | 80.7±0.6 | 73.91 |
| Galformer | 71.6±0.9 | 75.7±0.8 | 64.0±0.4 | 64.3±0.5 | 84.7±0.9 | 73.4±1.1 | 75.6±0.4 | 80.9±1.0 | 74.02 |
| GraphMVP | 71.5±1.3 | 76.1±0.9 | 64.3±0.6 | 64.7±0.7 | 85.4±0.8 | 74.9±1.2 | 76.0±0.6 | 81.5±1.2 | 74.86 |
| MoleBlend | 73.0±0.8 | 77.8±0.8 | 66.1±0.0 | 64.9±0.3 | 87.6±0.7 | 77.2±2.3 | 79.0±0.8 | 83.7±1.4 | 76.16 |
| LEMON | 73.7±1.1 | 77.5±0.6 | 65.1±0.5 | 64.3±0.9 | 85.9±3.2 | 79.4±4.3 | 79.3±1.1 | 87.8±1.4 | 76.62 |
| GEM | 72.4±0.4 | 78.1±0.1 | 69.2±0.4 | 67.2±0.4 | 90.1±1.3 | 81.7±0.5 | 80.6±0.9 | 85.6±1.1 | 78.18 |
| Uni-Mol | 72.9±0.6 | 79.6±0.5 | **69.6±0.1** | 65.9±1.3 | 91.9±1.8 | 82.1±1.3 | 80.8±0.3 | 85.7±0.2 | 78.63 |
| DeMol | **75.1±0.6** | **80.9±0.4** | 69.3±0.5 | **68.4±0.3** | **92.6±0.7** | **82.5±1.4** | **81.2±0.9** | **89.0±1.1** | **79.96** |

potential toxic effects, emphasising the importance of understanding molecular interactions at a biochemical level. Its diverse set of endpoints makes it a comprehensive resource for evaluating toxicity prediction models.

**ToxCast.** The ToxCast dataset is an extension of Tox21, providing a broader range of toxicity predictions. It includes over 10,000 molecules and covers more than 600 different toxicity assays, spanning various biological pathways and mechanisms. The dataset is designed to assess the potential adverse effects of chemicals on human health and the environment. ToxCast is particularly useful for evaluating models' ability to generalise across multiple toxicity endpoints, given its extensive coverage of biochemical interactions and diverse molecular structures.

**SIDER.** The ClinTox dataset is designed to predict clinical trial outcomes for drugs, specifically focusing on toxicity and efficacy. It contains 1,478 drugs, each labelled with two binary classifications: "toxic" or "non-toxic" during clinical trials, and "effective" or "ineffective" based on clinical results. ClinTox is derived from publicly available sources, including the DrugBank database and clinical trial records. The dataset is valuable for assessing models' ability to predict both safety and efficacy, which are critical factors in drug development.

**MUV.** The MUV dataset is a benchmark for virtual screening tasks, containing 150,000 molecules across 17 different biological targets. Each molecule is labelled as active or inactive for a specific target, representing a binary classification problem. MUV is designed to evaluate models' performance in identifying potential drug candidates by distinguishing active compounds from inactive ones. The dataset is particularly challenging due to its large size and the need to balance false positives and false negatives, making it a standard for benchmarking molecular property prediction models.

**HIV.** The HIV dataset is focused on predicting the activity of compounds against HIV (Human Immunodeficiency Virus). It contains 41,127 molecules, each labelled as active or inactive based on its ability to inhibit HIV replication. The dataset is derived from high-throughput screening experiments conducted by the Developmental Therapeutics Program (DTP) AIDS Antiviral Screen. HIV is a binary classification task that requires models to identify structural features associated with antiviral activity, making it essential for research in anti-HIV drug discovery.

**BACE.** The BACE (Beta-Secretase Cleavage Site) dataset is designed to predict the inhibition of the Beta-secretase enzyme, which plays a key role in Alzheimer's disease. It contains 1,513 molecules,

each labeled as active or inactive based on their ability to inhibit the enzyme. BACE is a binary classification task that evaluates models' ability to identify molecules with therapeutic potential for treating neurodegenerative diseases. The dataset is derived from experimental screens and is widely used to benchmark models in the context of enzyme inhibition and drug discovery.

**Settings.** In our experiments, referring to previous work Uni-Mol (Zhou et al., 2023), we use scaffold splitting to divide the dataset into training, validation, and test sets in the ratio of 8:1:1. Scaffold splitting is more challenging than random splitting as the scaffold sets of molecules in different subsets do not intersect. This splitting tests the model's generalization ability and reflects the realistic cases (Wu et al., 2018). In all experiments, we choose the checkpoint with the best validation loss, and report the results on the test-set run by that checkpoint. Referring to previous work, we use a grid search to find the best combination of hyperparameters for the molecular property tasks. To reduce the time cost, we set a smaller search space for the large datasets. The specific search space is shown in Table 7. For small dataset, we run them on a single NVIDIA A6000 GPU; for large datasets and HIV, we run them on 4 NVIDIA A6000 GPUs.

Table 7: Search space for small datasets: BBBP, BACE, ClinTox, Tox21, Toxcast, SIDER, for large datasets: MUV and HIV.

| Hyperparameter | Small | Large | HIV |
|---|---|---|---|
| Learning rate | [5e-5, 8e-5, 1e-4, 4e-4, 5e-4] | [2e-5, 1e-4] | [2e-5, 5e-5] |
| Batch size | [32, 64, 128, 256] | [128, 256] | [128, 256] |
| Epochs | [40 ,60, 80, 100] | [20, 40] | [2, 5, 10] |
| Pooler dropout | [0.0, 0.1, 0.2, 0.5] | [0.0, 0.1] | [0.0, 0.2] |
| Warmup ratio | [0.0, 0.06, 0.1] | [0.0, 0.06] | [0.0, 0.1] |

## E    VISUALISATION FOR SELF-ATTENTION MAP

For better interpretability, we conduct a visualisation on the self-attention map on atom-centric graph atom-atom attention and bond-centric graph bond-bond attention, respectively, as shown in Figure 5. The top section of Figure 5 illustrates the self-attention weights for the atom-centric graph representation. Each subfigure corresponds to one attention head, showing the pairwise attention weights between atoms in the molecule. The bottom section of Figure 5 depicts the self-attention weights for the bond-centric graph representation. Similar to the atom-centric case, each subfigure corresponds to one attention head, showing the pairwise attention weights between bonds in the molecule. By comparing the two sections of the figure, we observe that atom-centric attention tends to exhibit more varied and complex patterns, reflecting the rich diversity of atomic interactions and functional groups. Bond-centric attention, on the other hand, shows more structured and localised patterns, emphasising the importance of bond-specific features in capturing molecular properties. The model appears to effectively leverage both local and global structural information, as evidenced by the combination of diagonal and off-diagonal attention patterns. The complementary nature of these two representations suggests that combining atom-level and bond-level information can lead to a more comprehensive understanding of molecular structures.

## F    QUALITATIVE ANALYSIS OF BOND-LEVEL INTERACTIONS

We further conduct an experiment visualising the representation spaces of DeMol, GEM, and LEMON using a curated dataset of 100 molecules designed to test specific bond-level capabilities. We first curated a dataset containing four distinct classes of molecules to test two specific bond-level phenomena:

- Stereochemistry (3D Geometry): 25 cis-isomers (e.g., Cisplatin) and 25 trans-isomers (e.g., Transplatin). These differ only in 3D bond orientation (dihedral/bond angles).
- Resonance (2D Electronic Structure): 25 aromatic rings (e.g., Benzene) and 25 non-aromatic rings (e.g., Cyclohexadiene). These differ in bond order and delocalisation.

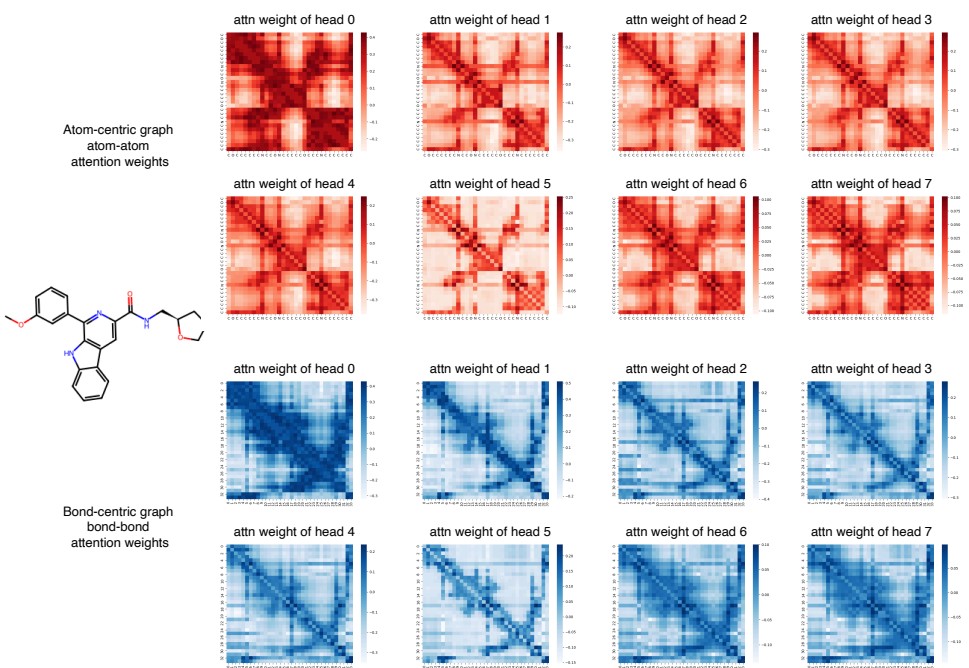

Figure 5: Visualisation on self-attention map of multi-heads independently.

We projected the final graph-level embeddings of these molecules using t-SNE. As shown in Figure 6, DeMol, GEM and LEMON successfully separate Aromatic (Green) from Non-Aromatic (Red) clusters, proving it captures 2D topological features well. However, the Cis- (Blue) and Trans- (Orange) isomers are completely mixed into a single cluster by LEMON. Although GEM explicitly models bond-angle graphs, its GNN-based message passing appears to "smooth" these features, leading to entangled representations where 3D geometric distinctions are not strictly enforced in the final embedding space. DeMol produces four distinct, well-separated clusters. It clearly distinguishes Cis- from Trans- isomers (capturing 3D geometry) and aromatic from non-aromatic rings (capturing bond resonance). This provides direct qualitative proof that DeMol's unique architecture successfully learns to "disentangle" complex bond-level phenomena that other bond-aware models either miss or conflate.

## G    LIMITATIONS AND FUTURE WORK

**Limitations.**    While DeMol explicitly models atom-bond and bond-bond interactions through dual-graph representations, the introduction of bond-centric channels and cross-level attention blocks increases computational overhead. We mitigate this problem by structure-aware masks derived from chemical valency rules, but further optimization is still required for real-time applications. Besides, the current framework is primarily validated on organic molecules and simple inorganic complexes. Its applicability to systems with unconventional bonding (e.g., metal-organic frameworks, organometallic compounds) or dynamic covalent interactions (e.g., reversible bonds in supramolecular assemblies) awaits further investigation.

**Future Work.**    In the future, we can explore lightweight cross-attention designs (e.g., sparse attention via learnable graph sparsification or kernelized approximations) to reduce computational costs while preserving geometric consistency. Moreover, it's promising to incorporate explicit electronic descriptors (e.g., partial charges, orbital hybridization) and quantum mechanical constraints (e.g., HOMO-LUMO gaps) into the bond-centric channel and explore hybrid models combining DeMol with physics-informed neural networks to bridge classical and quantum representations.

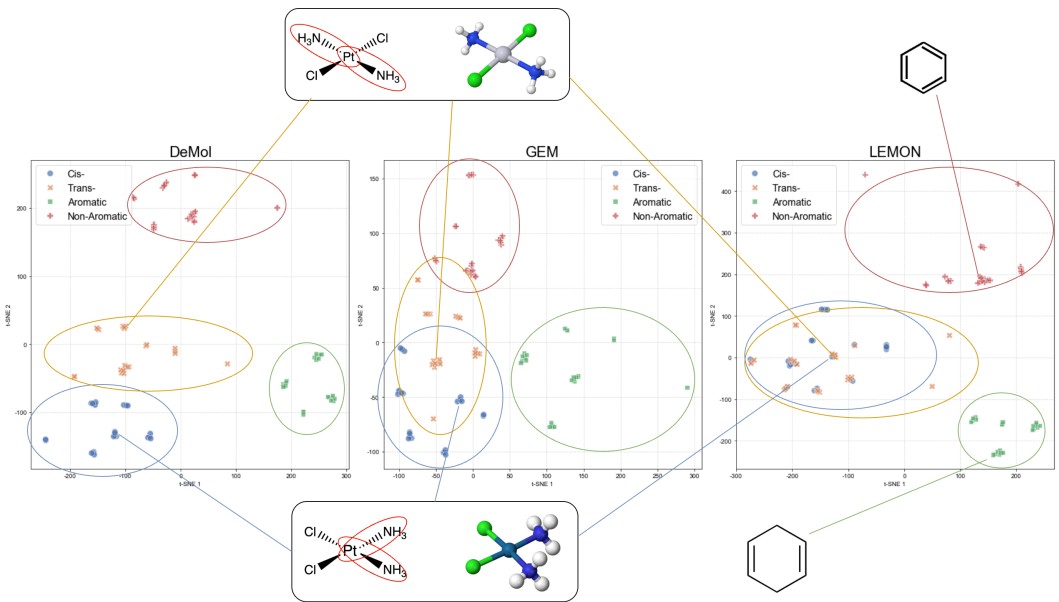

Figure 6: Visualisation on qualitative analysis of bond-level interactions.

## H  THE IMPACT OF PRE-TRAINING ON PCQM4Mv2

We conduct an additional experiment on the QM9 dataset. We train the DeMol without pretraining on the PCQM4Mv2 dataset. We selected five QM9 targets for comparison. All the hyperparameters of pre-training and fine-tuning are kept the same.

Table 8: The impact of pre-training on PCQM4Mv2.

| Methods | $\epsilon_{HOMO}$ | $\epsilon_{LUMO}$ | $\Delta\epsilon$ | $G$ | $C_v$ |
|---|---|---|---|---|---|
| No Pre-training | 21.2 | 19.3 | 30.4 | 10.97 | 0.029 |
| Pre-training on PCQM4Mv2 | 16.4 | 15.7 | 26.8 | 6.29 | 0.021 |

As shown in the Table 8, pretraining on the PCQM4Mv2 dataset does indeed improve performance on downstream tasks, consistent with the performance gains observed in Graphormer (Ying et al., 2021), Transformer-M (Luo et al., 2022) and Uni-mol (Zhou et al., 2023).

## I  AN EXAMPLE OF THE TWO GRAPH REPRESENTATIONS

Figure 7 illustrates two distinct graph representations of the methane molecule ($CH_4$), highlighting the complementary nature of atom-centric and bond-centric graph formulations. The figure is structured to provide a comprehensive view of how these representations encode structural information, including interatomic distances and inter-bond angles. The leftmost panel shows the chemical formula ($CH_4$) and its 2D and 3D structures, providing a visual reference for the methane molecule.

In the atom-centric graph, atoms are represented as nodes ($v_1, v_2, v_3, v_4, v_5$), with edges ($e_1, e_2, e_3, e_4$) connecting them to reflect the molecular connectivity. This representation explicitly encodes structural information such as interatomic distances. The structure encoding includes adjacency matrix, shortest path distance encodings, 3D distance encodings and so on. The adjacency matrix provides a numerical representation of the graph's connectivity. For example, the entry "1" indicates a direct connection between two atoms, while "0" signifies no direct connection. A 2D graph visualisation depicts the shortest path distances between atoms, emphasising the spatial relationships within the molecule. Nodes are connected by edges whose lengths reflect the shortest paths between them. On the far right, a 3D molecular model of methane is shown, with bonds highlighted in red.

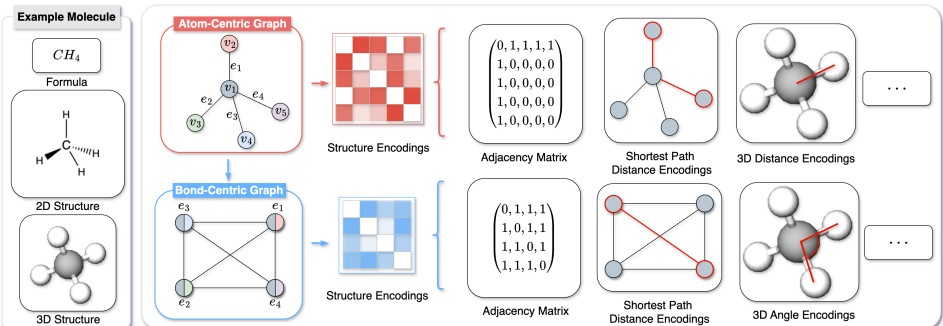

Figure 7: **An example of the two graph representations of the methane molecule. Atom-Centric Graph**: Graph representation with atoms as nodes, which can explicitly encode structural information such as interatomic distances. **Bond-Centric Graph**: Graph representation with bonds as nodes, which can explicitly encode structural information such as inter-bond angles.

This visualisation integrates geometric information, illustrating how the atom-centric graph can capture three-dimensional spatial relationships.

The bond-centric graph shifts the focus from atoms to bonds, where bonds are treated as nodes $(e_1, e_2, e_3, e_4)$. Edges between these nodes represent the relationships between adjacent bonds, enabling the explicit encoding of structural features such as inter-bond angles. Similar to the atom-centric graph, a colour-coded matrix (Structure Encodings) is provided to visualise the adjacency relationships between bonds. The matrix captures the connectivity patterns among bonds, reflecting their spatial arrangement. The adjacency matrix for the bond-centric graph is presented. This matrix numerically encodes the connectivity between bonds, with "1" indicating a direct relationship and "0" indicating no direct relationship. A 2D graph visualisation shows the shortest path distances between bonds, emphasising the topological relationships within the bond-centric graph. On the far right, a 3D molecular model of methane is again depicted, but this time with a focus on the angles between bonds. Bonds are highlighted in red, and the visualisation emphasises how the bond-centric graph can capture angular relationships in three-dimensional space.

## J   LLM USAGE DISCLOSURE

The authors utilized a large language model (LLM) solely as a tool to assist with the polishing and refinement of the writing in this paper. The model was used exclusively for improving grammatical fluency, sentence structure, and overall clarity of the manuscript. All ideation, theoretical development, empirical research, and technical conclusions remain entirely the work of the authors. The authors take full responsibility for all content generated by the LLM and presented in this work.

