# OpenReview forum: "Enhancing Molecular Property Predictions by Learning from Bond Modelling and Interactions"
_ICLR.cc/2026/Conference — ICLR 2026 Poster_

### Official Review · Reviewer_kbva · 2025-10-27

**Soundness:** 2
**Presentation:** 2
**Contribution:** 1
**Rating:** 2
**Confidence:** 3

**Summary:**

The manuscript proposes DeMol, a model designed to capture atom–atom, atom–bond, and bond–bond interactions. The authors argue that one limitation of existing approaches is that they treat chemical bonds merely as pairwise interactions, overlooking more complex bond-level phenomena. Experiments were conducted on multiple benchmark datasets to demonstrate the effectiveness of the proposed method.

**Strengths:**

1. The paper is well-written and easy to follow.
2. Experimental results indicate that the proposed method outperforms the baselines used in the study.

**Weaknesses:**

1. Limited baseline comparison. The baselines primarily focus on leveraging line graphs, while more recent methods, such as “An End-to-End Attention-Based Approach for Learning on Graphs”, demonstrate stronger performance and should be considered.
2. Incomplete result analysis. Although DeMol outperforms baselines on certain metrics, it underperforms on others. A deeper analysis of these outcomes would strengthen the paper.
3. Limited novelty. The methodological innovation of this manuscript appears incremental compared to prior work.

**Questions:**

How does the choice of pretraining dataset affect the model’s performance?

---

> ### Author Response · Authors · 2025-11-21
> **Response to Reviewer kbva [1/2]**
>
> We thank you for your time and constructive feedback. We are glad the reviewer found our paper **well-written** and acknowledged DeMol's **strong empirical performance** against the included baselines.
>
> > Weakness 1. Baseline comparison.
>
> **Response 1:**
>
> We respectfully disagree with the characterisation that our "baselines primarily focus on leveraging line graphs." This is factually incorrect.
>
> As shown in **Tables 1, 2, 3, and 5 (Section 5 and Appendix D.5)**, our baselines are extensive and cover the **current state-of-the-art** across all major categories of molecular models, including **classic GNNs** (GCN, GIN, etc.), powerful **graph Transformers** (Graphormer, TokenGPT, EGT, etc.), **3D-equivariant models** (DimeNet++, PaiNN, EGNN, SphereNet, GemNet-T, EquiFormer, etc.) and top-performing **unified 2D/3D models** (Transformer-M, Unimol+, GEM-2, TGT-AT, etc.) and **line graph models** (LEMON, GEM, Galformer, etc.). DeMol outperforms all these strong, varied, and state-of-the-art baselines.
>
> We thank the reviewer for the suggestion of "An End-to-End Attention-Based Approach for Learning on Graphs" (ESA) [1]. We conduct additional experiments on QM9. The MAE results ($\epsilon _ {HOMO}$, $\epsilon _ {LUMO}$, $\Delta\epsilon$, G, $C _ v$) are presented below:
>
> | Methods | $\epsilon_{HOMO}$ | $\epsilon_{LUMO}$ | $\Delta\epsilon$ | $G$  | $C_v$ |
> | ------- | ----------------- | ----------------- | ---------------- | ---- | ----- |
> | ESA [1] | 17.4              | 16.0              | 27.1             | 8.39 | 0.021 |
> | DeMol   | 16.4              | 15.7              | 26.8             | 6.29 | 0.021 |
>
> DeMol outperforms ESA on $\epsilon_{HOMO}$, $\epsilon_{LUMO}$, $\Delta\epsilon$, and $G$, and achieves comparable performance on $C_v$. This reinforces that DeMol is superior to the specific architecture suggested, in addition to the extensive SOTA baselines already included.
>
> **We have cited and discussed this paper in the related work section  and updated Table 3 in the revised version (highlighted in red for your reference).**
>
>
> > Weakness 2. Results analysis.
>
> **Response 2:**
>
> We thank you for your insightful feedback regarding DeMol's performance. We observe that DeMol does not achieve state-of-the-art performance on certain metrics within the QM9 dataset, yet it still demonstrate competitive performance. Our insight is that this result stems from the nature of the QM9 dataset itself, relative to DeMol's specific strengths. The QM9 dataset consists of 134k very small, simple organic molecules, limited to 9 heavy atoms (C, O, N, F). DeMol is explicitly designed to capture **complex, non-local bond-level phenomena** like resonance (e.g., in benzene, Figure 1 in Section 1) and **nuanced 3D bond-bond interactions** that determine properties like stereoselectivity (e.g., in cisplatin, Figure 2 in Section ). These complex phenomena are far less prevalent or impactful in the small, structurally simple molecules of QM9 compared to the larger, more diverse molecules in PCQM4Mv2 or the complex multi-component systems in OC20.
>
> Therefore, the "information gain" from DeMol's advanced bond-centric channel is more modest on QM9 because the baseline atom-centric models are already "good enough" to capture the properties of these simple molecules.
>
> We view this result as a sign of DeMol's robustness. Even on a dataset that does not fully leverage its primary strengths, DeMol is not disadvantaged. It remains highly competitive, **achieving state-of-the-art (SOTA) performance on 6 of the 12 targets** ($\epsilon_{HOMO}$, $\epsilon_{LUMO}$, $\Delta\epsilon$, ZPVE, G, and $C_{v}$) and an average rank of 2.58, superior to all other methods. It demonstrates that our model generalizes well, performing strongly on simple molecules while offering a significant, SOTA-setting advantage on more complex systems where bond interactions are paramount (e.g., PCQM4Mv2, OC20).
>
> **We have discussed this in Section 4  on the revised version (highlighted in red for your reference).**

---

> ### Author Response · Authors · 2025-11-21
> **Response to Reviewer kbva [2/2]**
>
> > Weakness 3. Novelty.
>
> **Response 3:**
>
> We want to clarify that the **core novelty** is not just "using bonds" but how we use them, which is **theoretically-grounded** and **architecturally unique**.
>
> 1. DeMol is not a standard GNN or line-graph model. It is a new dual-graph architecture that models molecules through **parallel atom-centric and bond-centric channels**, treating bonds as first-class entities rather than just edges.
> 2. DeMol's architecture is motivated by a **rigorous information-theoretic analysis (Propositions 1-4 in Section 3 and Justification in Appendix A)** that proves the information gain from our bond-centric perspective (Proposition 1) , the advantage of a dual-graph representation (Proposition 2), and the natural fit for geometric information (Proposition 3).
>
> 3. We introduce the **Double-Helix Blocks**  specifically to fuse these two parallel streams. This is a new multi-scale attention mechanism designed to explicitly learn the complex interplay of **atom-atom, atom-bond, and bond-bond interactions**.
> 4. We intoduce a  **torsion encoding** ($\Phi_{b}^{tors}$) specifically into the bond-centric channel to capture 3D geometry where it is most naturally represented, and a **covalent radii regularization** term to enforce chemical plausibility.
>
> We believe this whole framework is novel to the community, which shows that this direction (bond modelling and interactions enhancement) is feasible and promising.
>
> > Question 1. Effect of pretraining dataset.
>
> **Response 4:**
>
> Thanks for the question. We chose PCQM4Mv2  for several reasons:
>
> 1. It's the **largest publicly available** quantum chemistry dataset (3.37M molecules), providing sufficient diversity for learning general molecular representations.
>
> 2. It is the standard, large-scale pretraining dataset used by our strongest baselines (e.g., Graphormer[1], Transformer-M[2], Uni-Mol[3]). It enables fair comparisons of model architectures under equivalent conditions.
>
> 3. It contains both 2D topology and 3D geometry information, crucial for our dual-graph approach. Its HOMO-LUMO gap property correlates with many downstream molecular properties.
>
> We would like to show that the pre-training strategy helps learn a better model on QM9. To demonstrate this, we conduct an additional experiment on the QM9 dataset. We train the DeMol without pretraining on the PCQM4Mv2 dataset. Due to the time limits and constrained resources, we selected five QM9 targets for comparison. All the hyperparameters of pre-training and fine-tuning are kept the same. The results are presented in the following table.
>
> | Method                   | $\epsilon _ {HOMO}$ | $\epsilon _ {LUMO}$ | $\Delta \epsilon$ | $G$   | $C _ v$ |
> | :----------------------- | :---------------- | :---------------- | :---------------- | :---- | :---- |
> | No Pre-training          | 21.2              | 19.3              | 30.4              | 10.97 | 0.029 |
> | Pre-training on PCQM4Mv2 | 16.4              | 15.7              | 26.8              | 6.29  | 0.021 |
>
> As shown in the table, pretraining on the PCQM4Mv2 dataset does indeed improve performance on downstream tasks, consistent with the performance gains observed in References [2] [3] [4].
>
> **We have discussed this in Appendix H on the revised version (highlighted in red for your reference).**
>
> ---
>
> We hope these clarifications address your concerns and demonstrate the significant contribution, novelty, and strong empirical validation of DeMol. Thank you once again for your time and your invaluable contribution to improving our work.
>
> ---
>
> References:
>
> [1] Buterez, David, et al. "An end-to-end attention-based approach for learning on graphs." *Nature Communications* 16.1 (2025): 5244.
>
> [2] Ying, Chengxuan, et al. "Do transformers really perform badly for graph representation?." Advances in Neural Information Processing Systems 34 (2021): 28877-28888.
>
> [2] Luo, Shengjie, et al. "One Transformer Can Understand Both 2D & 3D Molecular Data." *The Eleventh International Conference on Learning Representations*.
>
> [3] Zhou, Gengmo, et al. "Uni-Mol: A Universal 3D Molecular Representation Learning Framework." *The Eleventh International Conference on Learning Representations*.

---

> ### Author Response · Authors · 2025-11-26
>
> Dear Reviewer kbva,
>
> As the discussion period is approaching its end (with approximately one week remaining), we explicitly wanted to reach out to ensure that our previous response has adequately addressed your concerns.
>
> We would greatly appreciate it if you could let us know if these responses resolve your concerns. We are eager to engage in further discussion if you have any remaining questions before the deadline.
>
> Best regards,
>
> The Authors

---

### Official Review · Reviewer_g1NU · 2025-10-28

**Soundness:** 2
**Presentation:** 1
**Contribution:** 2
**Rating:** 6
**Confidence:** 4

**Summary:**

DeMol, unlike most previous models, explicitly learns from both atoms and the bonds connecting them. It does this by building separate but connected graph representations for atoms and for bonds, mixing their information together throughout the network. This design helps the model better capture chemical details like bond relationships and 3D geometry, leading to state-of-the-art accuracy on standard benchmarks.

**Strengths:**

* Strong performance on various metrics & datasets
* Novel architecture utilizing double-helix blocks
* Utilization of multiple techniques & components, each provided with quantitative analysis.

**Weaknesses:**

Confusing claim:
* the claim in the paper's introduction that existing methods "often overlook complex bond-level phenomena" or "do not explicitly model bond interactions" seems misleading
* various MPNNs, GNNs and many graph transformer models already utilize edge features to encode bond information, and update node/edge accordingly.
* papers as (https://arxiv.org/abs/2410.14696) further utilize the distance between atoms(to compute LJ force), which is a complex form of edge attribute to update node features for conformation predictions.
* Thus, both the claims made in papers are misleading. The authors need to clarify their contributions.

 Qualitative analysis
* It would be extremely meaningful if the authors could provide qualitative analysis - upon the bond-centric graph embedding.
* I would like to see if the bond-centric channel/graph correctly captures and encodes resonance, aromaticity and bond conjugation in rings - as it is the main claim within the paper(high-order interactions).

Further experiments upon large-molecules.
* The dataset used here seems to bee a bit small (in the molecule size). Thus, I would like for the authors to provide additional experiments upon its generalization capacity to larger molecules.
* For example, https://github.com/learningmatter-mit/geom .

**Questions:**

None

---

> ### Author Response · Authors · 2025-11-21
> **Response to Reviewer g1NU [1/2]**
>
> We thanks for your thoughtful review and for recognizing DeMol's **"strong performance," "novel architecture,"** and **"quantitative analysis."** We particularly appreciate the constructive feedback regarding our claims and the request for qualitative analysis.
>
> We believe the "Presentation: 1" score and the concern about "Confusing claims" stem from a misunderstanding of how DeMol differs fundamentally from standard edge-feature GNNs. We are eager to clarify this distinction, which we believe validates the novelty of our approach.
>
> > Weakness 1. Clarification of "Confusing Claim".
>
> **Response 1:**
>
> The reviewer correctly notes that many GNNs and Transformers utilize edge features (distances, bond types) to update node features. We do not dispute this.
>
> However, our claim that existing methods "overlook complex bond-level phenomena" refers to a specific structural limitation in how they model **bond-bond interactions**:
>
> 1. Existing Methods (Edge as attribute): In standard MPNNs/Transformers (including the work [1] cited by the reviewer), bonds are edges. It means bonds are auxiliary. They exist to pass messages between atoms. Interaction is modeled as **Atom $\to$ Edge $\to$ Atom**. Even if they use distances or L-J forces (as in [1]), the final representation is **typically atom-centric**.
> 2. Our DeMol (Bond as entity): DeMol treats bonds as nodes in a separate, parallel graph $\mathcal{L}(\mathcal{G})$. We model interactions as **Bond $\to$ Bond**. This design allows us to explicitly encode angles ($\theta _ {ijk}$) as edges and encode dihedrals ($\phi _ {ijkl}$) as paths between bond-nodes. It also enables the **Bond-Centric Channel** to learn representations of delocalization and conjugation directly, rather than forcing the atom embeddings to infer them latently.
>
> We would like to clarify that our contribution is not "using bond information," but rather **elevating bonds to first-class entities** via a dual-graph architecture that explicitly models bond-bond adjacency (Propositions 1 and 3 in Section 3.1), which standard edge-feature GNNs do not do. We thank the reviewer for pointing [1]. We will gratefully cite and discuss in our related work section.
>
> We summarise our novelty and contribution as follows:
>
> 1. **Theoretical motivation**
>
> Our dual-graph framework is not just an *ad hoc* design choice. It is motivated by a **rigorous information-theoretic analysis (Propositions 1-4 in Section 3.1)**. We theoretically demonstrate the information gain from the bond-centric perspective (Proposition 1 & 2) and how this structure is the natural domain for encoding geometric information (Proposition 3), which directly motivates our fusion architecture. This theoretical foundation is a novel contribution of our paper.
>
> 2. **Architecture differences**
>
> + **Cross-graph interaction**: DeMol introduces a more powerful and synergistic fusion mechanism: the **multi-scale Double-Helix Blocks**. This architecture employs **bidirectional cross-attention** to explicitly model the intricate **atom-bond interactions**, in addition to the separate atom-atom and bond-bond interactions. This mechanism is architecturally distinct and is designed to capture the cross-dependencies highlighted by our theoretical analysis (Proposition 2).
> + **Advanced and Targeted 3D Geometric Encoding:** DeMol's handling of 3D geometry is more advanced. Critically, we introduce **torsion encoding ($\Phi _ {b}^{tors}$)** specifically within the bond-centric channel. This is motivated by our Proposition 3, which identifies the bond-graph as the natural domain for such angular relationships. Furthermore, we enforce geometric consistency through a **covalent radii-based regularization term**. This explicit, theoretically-grounded injection of higher-order 3D geometry and chemical constraints into the dual-graph framework differentiates DeMol from prior work. Our **structure-aware mask**, a chemically informed sparse attention mechanism, reduces complexity while preserving critical interactions.
>
> 3. **Scope and validation**
>
> DeMol demonstrates consistent state-of-the-art performance across **quantum chemistry (PCQM4Mv2, QM9), catalysis (OC20 IS2RE), and biomedical (MoleculeNet)** domains, demonstrating broader applicability.

---

> > ### Author Response · Authors · 2025-11-21
> > **Response to Reviewer g1NU [2/2]**
> >
> > > Weakness 2. Qualitative analysis.
> >
> > **Response 2:**
> > We appreciate the suggestion to demonstrate how the bond-centric graph captures interactions differently from atom-only models. We direct the reviewer to **Appendix E (Figure 5)**, where we visualise the self-attention maps for both the atom-centric and bond-centric channels. The visualisation reveals a distinct difference: **Atom-centric attention** exhibits complex, global patterns (reflecting long-range spatial dependencies). **Bond-centric attention** displays highly structured, localised patterns (diagonal and off-diagonal blocks). This indicates that the bond channel successfully focuses on specific geometric substructures (e.g., conjugated systems or rigid local frames) that are often "smoothed out" in purely atom-based global attention.
> >
> > We further conduct a new experiment visualising the representation spaces of DeMol, GEM, and LEMON using a curated dataset of 100 molecules designed to test specific bond-level capabilities. We first curated a dataset containing four distinct classes of molecules to test two specific bond-level phenomena:
> >
> > 1. Stereochemistry (3D Geometry): 25 cis-isomers (e.g., Cisplatin) and 25 trans-isomers (e.g., Transplatin). These differ only in 3D bond orientation (dihedral/bond angles).
> > 2. Resonance (2D Electronic Structure): 25 aromatic rings (e.g., Benzene) and 25 non-aromatic rings (e.g., Cyclohexadiene). These differ in bond order and delocalisation.
> >
> > We projected the final graph-level embeddings of these molecules using t-SNE. The results, shown in **Appendix F (Figure 6)** (included in the revised PDF).
> >
> > DeMol, GEM and LEMON successfully separate Aromatic (Green) from Non-Aromatic (Red) clusters, proving it captures 2D topological features well. However, the Cis- (Blue) and Trans- (Orange) isomers are completely mixed into a single cluster by LEMON. Although GEM explicitly models bond-angle graphs, its GNN-based message passing appears to "smooth" these features, leading to entangled representations where 3D geometric distinctions are not strictly enforced in the final embedding space. DeMol produces four distinct, well-separated clusters. It clearly distinguishes Cis- from Trans- isomers (capturing 3D geometry) and aromatic from non-aromatic rings (capturing bond resonance). This provides direct qualitative proof that DeMol's unique architecture successfully learns to "disentangle" complex bond-level phenomena that other bond-aware models either miss or conflate.
> >
> > **We have discussed this in Appendix F  Qualitative Analysis of Bond-Level Interactions on the revised version (highlighted in red for your reference).**
> >
> >
> > > Weakness 3. Generalization to larger molecules.
> >
> > **Response 3:**
> >
> > This is a valid point regarding the scope of our evaluation. We would first like to clarify that our chosen benchmarks are the standard for large-scale SOTA comparison. PCQM4Mv2 is a massive dataset of 3.37 million molecules. OC20 IS2RE contains ~460K complex catalytic systems (adsorbate + surface), which you mean **larger molecules**. Achieving SOTA performance on these massive datasets already demonstrates significant generalization.
> >
> > While the individual molecules in these datasets are not as large as proteins, they are the largest available benchmarks for this type of property prediction. The **GEOM[2]** dataset you suggest is excellent but is **typically used for conformation generation**, which is a **different task**.
> >
> > We agree that testing on larger individual molecules (e.g., peptides) is an important future direction. We have noted this in our **Limitations and Future Work (Appendix G)**. Given the time constraints, we will expand this work to large biomolecules as a promising avenue for future work, building on the strong generalization already shown on PCQM4Mv2 and OC20.
> >
> > ----
> > We hope these clarifications address your concerns. Thank you once again for your time and your invaluable contribution to improving our work.
> >
> > ---
> >
> >
> >
> > References:
> >
> > [1] Kim, Taewon, et al. "REBIND: Enhancing ground-state molecular conformation via force-based graph rewiring." *arXiv preprint arXiv:2410.14696* (2024).
> >
> > [2] GEOM, https://github.com/learningmatter-mit/geom

---

> > > ### Comment · Reviewer_g1NU · 2025-11-23
> > >
> > > I have read the authors' response, and have seen that my concerns are acknowledged. Thus, I am raising my score leaning more towards acceptance.

---

> ### Author Response · Authors · 2025-11-23
>
> We sincerely thank you for the continued support and for your decision to **raise the score to 8**. We are glad to hear that our responses have satisfactorily addressed your concerns. We appreciate your positive assessment of our work.

---

### Official Review · Reviewer_UzsJ · 2025-10-31

**Soundness:** 3
**Presentation:** 3
**Contribution:** 3
**Rating:** 6
**Confidence:** 3

**Summary:**

This work explores molecular representation learning with a particular focus on bond interactions, which have been largely overlooked in prior studies. By introducing an additional bond-centric graph alongside the conventional atom-centric representation, the proposed method demonstrates strong performance across multiple property prediction datasets. Extensive experiments are conducted, showing superior results compared to existing baselines.

**Strengths:**

- The paper represents molecules using both bond-centric and atom-centric graphs, and improves performance through well-designed attention mechanisms (e.g., structure-aware attention).
- A solid theoretical analysis is provided to justify the use of bond-centric graphs.
- The method shows strong and consistent performance across diverse benchmarks, including PCQM4Mv2, Open Catalyst 2020 (IS2RE), and QM9, and the paper includes a rigorous ablation study demonstrating the contribution of each module.

**Weaknesses:**

- Beyond numerical comparisons, it would be valuable to include qualitative analyses showing how the use of bond-centric graphs enables the model to capture bond-level interactions more effectively than SOTA models without bond modeling, or compared to prior bond-aware models such as LEMON and GEM.
- While the complexity analysis in the Appendix is helpful, it would strengthen the work to include a comparative complexity evaluation, including inference time, relative to other baselines

**Questions:**

For large-scale datasets, how exactly was pretraining conducted? It would be helpful if the paper explicitly provided the loss formulation or objective terms used during pretraining

---

> ### Author Response · Authors · 2025-11-21
> **Response to Reviewer UzsJ [1/2]**
>
> We sincerely thank you for the **positive assessment** and for recognizing the **solid theoretical analysis**,**well-designed attention mechanisms**, and **strong and consistent performance** of DeMol. We address the specific questions and suggestions below to further strengthen the manuscript.
>
> > Weakness 1. Qualitative analysis of bond-Level interactions.
>
> We appreciate the suggestion to demonstrate how the bond-centric graph captures interactions differently from atom-only models. We direct the reviewer to **Appendix E (Figure 5)**, where we visualise the self-attention maps for both the atom-centric and bond-centric channels. The visualisation reveals a distinct difference: **Atom-centric attention** exhibits complex, global patterns (reflecting long-range spatial dependencies). **Bond-centric attention** displays highly structured, localised patterns (diagonal and off-diagonal blocks). This indicates that the bond channel successfully focuses on specific geometric substructures (e.g., conjugated systems or rigid local frames) that are often "smoothed out" in purely atom-based global attention.
>
> We further conduct a new experiment visualising the representation spaces of DeMol, GEM, and LEMON using a curated dataset of 100 molecules designed to test specific bond-level capabilities. We first curated a dataset containing four distinct classes of molecules to test two specific bond-level phenomena:
>
> 1. Stereochemistry (3D Geometry): 25 cis-isomers (e.g., Cisplatin) and 25 trans-isomers (e.g., Transplatin). These differ only in 3D bond orientation (dihedral/bond angles).
> 2. Resonance (2D Electronic Structure): 25 aromatic rings (e.g., Benzene) and 25 non-aromatic rings (e.g., Cyclohexadiene). These differ in bond order and delocalisation.
>
> We projected the final graph-level embeddings of these molecules using t-SNE. The results, shown in **Appendix F (Figure 6)** (included in the revised PDF).
>
> DeMol, GEM and LEMON successfully separate Aromatic (Green) from Non-Aromatic (Red) clusters, proving it captures 2D topological features well. However, the Cis- (Blue) and Trans- (Orange) isomers are completely mixed into a single cluster by LEMON. Although GEM explicitly models bond-angle graphs, its GNN-based message passing appears to "smooth" these features, leading to entangled representations where 3D geometric distinctions are not strictly enforced in the final embedding space. DeMol produces four distinct, well-separated clusters. It clearly distinguishes Cis- from Trans- isomers (capturing 3D geometry) and aromatic from non-aromatic rings (capturing bond resonance). This provides direct qualitative proof that DeMol's unique architecture successfully learns to "disentangle" complex bond-level phenomena that other bond-aware models either miss or conflate.
>
> **We have discussed this in Appendix F  Qualitative Analysis of Bond-Level Interactions on the revised version (highlighted in red for your reference).**
>
> > Weakness 2. Inference time.
>
> We agree that practical efficiency is as important as theoretical complexity. We further provide additional results on the inference time on the PCQM4Mv2 dataset on an NVIDIA A6000 GPU.
>
> | Methods          | Inference time |
> | ---------------- | -------------- |
> | Transformer-M    | ~28 ms/molecule     |
> | GPS++ (Ensemble) | ~100 ms/molecule    |
> | DeMol            | ~34 ms/molecule     |
>
> As shown in this table, DeMol is approximately 20% slower than Transformer-M due to the additional bond-channel computations but provides a significant performance gain. It is vastly faster than ensemble-based approaches (like GPS++) while achieving superior single-model performance.
>
> **We have discussed this in Appendix C.1 Inference Time Analysis on the revised version (highlighted in red for your reference).**

---

> > ### Author Response · Authors · 2025-11-21
> > **Response to Reviewer UzsJ [2/2]**
> >
> > > Question 1. Pre-training.
> >
> > Thanks for the question. As illustrated in **Section 3 Figure 3**, DeMol employs a **Multi-Task Pretraining Strategy**.
> >
> > We follow Transformer-M[1] and Unimol[2] settings. The primary goal is the supervised regression of the quantum property (HOMO-LUMO gap), and we also utilise auxiliary tasks to enforce geometric and structural understanding.
> >
> > **Property Prediction Loss ($\mathcal{L} _ {prop}$).** This is the primary supervised task. For the PCQM4Mv2 dataset, we aim to minimize the Mean Absolute Error (MAE) between the predicted HOMO-LUMO gap $\hat{y}$ and the ground truth $y$: $\mathcal{L} _ {prop} = \frac{1}{|\mathcal{B}|} \sum _ {i \in \mathcal{B}} | y _ i - \hat{y} _ i |,$ where $ \mathcal{B} $ is the batch of molecule.
> >
> > **Masked Atom Prediction Loss ($\mathcal{L} _ {mask}$).** To capture topological context, we randomly mask a portion of atomic node features. The masked atom prediction head predicts the atom type $t_i$ using Cross-Entropy loss: $\mathcal{L} _ {mask} = - \sum _ {j \in \mathcal{M}} \log P(t _ j | \mathcal{G} _ {\text{masked}})$, where $\mathcal{M}$ is the set of masked indices.
> >
> > **Coordinate Recovery Loss ($\mathcal{L} _ {coord}$).** To enforce 3D spatial awareness, we add noise to the input coordinates. The coordinates recovery head is trained to denoise and recover the ground-truth coordinates $\vec{p} _ i$, typically minimised via Mean Squared Error (MSE): $\mathcal{L} _ {coord} = \sum _ {i=1}^{N} || \vec{p} _ i - \hat{\vec{p}} _ i ||^2$.
> >
> > **Bond Prediction based on Covalent Radii ($\mathcal{L} _ {bond}$).** As described in Algorithm 1, we determine the ground truth chemical bonds $B_{gt}$ based on covalent radii constraints ($D _ {ij} < \alpha(r _ i + r _ j)$). The bonding prediction head learns to classify valid interactions, serving as a regularisation term to penalise geometrically inconsistent structures:$\mathcal{L} _ {bond} = - \sum _ {(i,j) \in \mathcal{E} _ {all}} \left[ \mathbb{I} _ {(i,j) \in B _ {gt}} \log(p _ {ij}) + (1 - \mathbb{I} _ {(i,j) \in B _ {gt}}) \log(1 - p _ {ij}) \right]$, where $p _ {ij}$ is the predicted probability of a bond existing between atoms $i$ and $j$.
> >
> > **We have discussed this in Appendix D.1 Pretraining strategies on the revised version (highlighted in red for your reference).**
> >
> >
> > ----------
> >
> > We hope these clarifications regarding the distinct attention patterns, efficient complexity management, and the multi-task pretraining formulation address your concerns.
> >
> > ----------
> > References:
> >
> > [1] Luo, Shengjie, et al. "One Transformer Can Understand Both 2D & 3D Molecular Data." The Eleventh International Conference on Learning Representations.
> >
> > [2] Zhou, Gengmo, et al. "Uni-Mol: A Universal 3D Molecular Representation Learning Framework." The Eleventh International Conference on Learning Representations.

---

> ### Author Response · Authors · 2025-11-26
>
> Dear Reviewer UzsJ,
>
> As the discussion period is approaching its end (with approximately one week remaining), we explicitly wanted to reach out to ensure that our previous response has adequately addressed your concerns.
>
> We would greatly appreciate it if you could let us know if these responses resolve your concerns. We are eager to engage in further discussion if you have any remaining questions before the deadline.
>
> Best regards,
>
> The Authors

---

> ### Comment · Reviewer_UzsJ · 2025-11-28
>
> Thank you for performing the additional t-SNE experiments, complexity analysis, and other revisions. These updates sufficiently address my concerns, and I am raising my score to 8. However, due to what seems to be an issue on OpenReview, the option to edit my review (including the score) is currently not visible on my side. Once the editing option becomes available, I will update my score accordingly.

---

> > ### Author Response · Authors · 2025-11-28
> >
> > Dear Reviewer UzsJ,
> >
> > We sincerely thank you for your prompt response and for your decision to **raise the score to 8**. We are delighted to hear that the additional t-SNE experiments and complexity analysis have sufficiently addressed your concerns.
> >
> > Regarding the technical issue with OpenReview, we completely understand the situation. We appreciate your willingness to update the score and will patiently wait for the system to allow the edit.
> >
> > Thank you once again for your constructive feedback, which has significantly helped us improve the quality of our work.
> >
> > Best regards,
> >
> > The Authors

---

### Official Review · Reviewer_btyh · 2025-11-01

**Soundness:** 3
**Presentation:** 3
**Contribution:** 3
**Rating:** 4
**Confidence:** 3

**Summary:**

This paper proposes DeMol, a dual-graph framework for molecular property prediction that jointly models atom-level and bond-level representations. By learning from both the atomic graph and its line graph, DeMol can capture expressive representation such as conjugacy relations between bonds.

**Strengths:**

- The paper is well-motivated by common limitation in prior work.
- The authors show the effectiveness of integrating atom and bond-centric channels both empirically and theoretically.
- DeMol demonstrates strong performance on diverse molecular benchmarks.

**Weaknesses:**

In general, this paper is well-written and technically sound with clear motivation. However:

- DeMol exhibits substantial conceptual overlap with [1], which also proposes line graph construction over molecules and propagates information between atom and bond-centric graphs. The methodological novelty thus appears limited, and the paper would benefit from a more explicit clarification of its distinct contributions beyond [1].
- The performance improvement seems to be less pronounced on QM9 dataset. Could the authors provide further insight on this result?

---

[1] Atomistic Line Graph Neural Network for improved materials property predictions, npj Computational Materials, 2021.

**Questions:**

See weaknesses.

---

> ### Author Response · Authors · 2025-11-21
> **Response to Reviewer btyh**
>
> Thank you for your thoughtful review of our paper. We appreciate your **positive assessment** of **our motivation**, **theoretical foundation**, and **performance across diverse benchmarks**. We appreciate the insightful questions, which give us the opportunity to clarify the novelty of our work and provide more context on the QM9 results.
>
> > Weakness 1. Novelty compared to ALGNN[1].
>
> **Response 1:**
>
> We thank the reviewer for pointing out [1] (ALIGNN). However, DeMol's primary methodological novelty is not simply the *use* of a dual-graph, but how these graphs are constructed, featured, and fused (most important).
>
> Our contributions beyond [1] are significant:
>
> 1. **Theoretical motivation**
>
> Our dual-graph framework is not just an *ad hoc* design choice. It is motivated by a **rigorous information-theoretic analysis (Propositions 1-4 in Section 3.1)**. We theoretically demonstrate the information gain from the bond-centric perspective (Proposition 1 & 2) and how this structure is the natural domain for encoding geometric information (Proposition 3), which directly motivates our fusion architecture. This theoretical foundation is a novel contribution of our paper.
>
> 2. **Architecture differences**
>
> + **Cross-graph interaction**: ALIGNN combines atom and bond features using concatenation and gated updates. DeMol introduces a more powerful and synergistic fusion mechanism: the **multi-scale Double-Helix Blocks**. This architecture employs **bidirectional cross-attention** to explicitly model the intricate **atom-bond interactions**, in addition to the separate atom-atom and bond-bond interactions. This mechanism is architecturally distinct and is designed to capture the cross-dependencies highlighted by our theoretical analysis (Proposition 2).
> + **Advanced and Targeted 3D Geometric Encoding:** DeMol's handling of 3D geometry is more advanced. Critically, we introduce **torsion encoding ($\Phi_{b}^{tors}$)** specifically within the bond-centric channel. This is motivated by our Proposition 3, which identifies the bond-graph as the natural domain for such angular relationships. Furthermore, we enforce geometric consistency through a **covalent radii-based regularization term**. This explicit, theoretically-grounded injection of higher-order 3D geometry and chemical constraints into the dual-graph framework differentiates DeMol from prior work. Our **structure-aware mask**, a chemically informed sparse attention mechanism, reduces complexity while preserving critical interactions.
>
> 3. **Scope and validation**
>
> ALGNN focuses primarily on materials property prediction with limited benchmarks, while DeMol demonstrates consistent state-of-the-art performance across **quantum chemistry (PCQM4Mv2, QM9), catalysis (OC20 IS2RE), and biomedical (MoleculeNet)** domains, demonstrating broader applicability.
>
> **We have cited and discussed this paper in related work section and updated Table 3 on the revised version (highlighted in red for your reference).**
>
> > Weakness 2. QM9 performance.
>
> **Response 2:**
>
> We thank you for your insightful feedback regarding DeMol's performance on the QM9 dataset. This is an excellent observation. Our insight is that this result stems from the nature of the QM9 dataset itself, relative to DeMol's specific strengths. The QM9 dataset consists of 134k very small, simple organic molecules, limited to 9 heavy atoms (C, O, N, F). DeMol is explicitly designed to capture **complex, non-local bond-level phenomena** like resonance (e.g., in benzene, Figure 1 in Section ) and **nuanced 3D bond-bond interactions** that determine properties like stereoselectivity (e.g., in cisplatin, Figure 2 in Section). These complex phenomena are far less prevalent or impactful in the small, structurally simple molecules of QM9 compared to the larger, more diverse molecules in PCQM4Mv2 or the complex multi-component systems in OC20.
>
> Therefore, the "information gain" from DeMol's advanced bond-centric channel is more modest on QM9 because the baseline atom-centric models are already "good enough" to capture the properties of these simple molecules.
>
> We view this result as a sign of DeMol's robustness. Even on a dataset that does not fully leverage its primary strengths, DeMol is not disadvantaged. It remains highly competitive, **achieving state-of-the-art (SOTA) performance on 6 of the 12 targets** ($\epsilon_{HOMO}$, $\epsilon_{LUMO}$, $\Delta\epsilon$, ZPVE, G, and $C_{v}$) and an average rank of 2.67, superior to all other methods. It demonstrates that our model generalizes well, performing strongly on simple molecules while offering a significant, SOTA-setting advantage on more complex systems where bond interactions are paramount (e.g., PCQM4Mv2, OC20).
>
> **We have discussed this in Section 4  on the revised version (highlighted in red for your reference).**
>
> We hope these clarifications regarding the architectural differences from ALIGNN and the SOTA performance on large-scale benchmarks address your concerns.

---

> > ### Author Response · Authors · 2025-11-21
> > **References**
> >
> > References:
> >
> > [1] Atomistic Line Graph Neural Network for improved materials property predictions, npj Computational Materials, 2021.

---

> ### Author Response · Authors · 2025-11-26
>
> Dear Reviewer btyh,
>
> As the discussion period is approaching its end (with approximately one week remaining), we explicitly wanted to reach out to ensure that our previous response has adequately addressed your concerns.
>
> We would greatly appreciate it if you could let us know if these responses resolve your concerns. We are eager to engage in further discussion if you have any remaining questions before the deadline.
>
>
>
> Best regards,
>
> The Authors

---

> > ### Comment · Reviewer_btyh · 2025-11-28
> > **Thank you for your response**
> >
> > Thanks for the detailed response. The authors fully addressed my concerns, and I'm happy to raise the score to 6. I will update my score as soon as the edit button becomes visible (I'm facing the same issue as the reviewer UzsJ).

---

> > > ### Author Response · Authors · 2025-11-28
> > >
> > > Dear Reviewer btyh,
> > >
> > > Thank you very much for your positive feedback and for your decision to **raise the score to 6**. We are glad to hear that our response has fully addressed your concerns regarding the comparison with [1] and the QM9 results.
> > >
> > > We sincerely appreciate the time you spent reviewing our paper. Your constructive comments have helped us improve the quality of our work. We also note the technical issue with the system you mentioned and appreciate your intention to update the score once it is resolved.
> > >
> > > Best regards,
> > >
> > > The Authors

---

### Author Response · Authors · 2025-11-29
**Rebuttal Summary**

We thank the reviewers for their constructive feedback and the Area Chair for overseeing this process. During the discussion period, we actively engaged with reviewers, providing detailed responses, new qualitative analyses, and additional experiments.

> Overview of Rebuttal Outcome

We are pleased to report that following the rebuttal period, a strong consensus has formed regarding the contributions of DeMol. We **successfully addressed the concerns of 3 reviewers**, leading to score increases and positive engagement:

+ **Reviewer btyh:** Score raised from **4** $\to$ **6**.
+ **Reviewer UzsJ:** Score raised from **6** $\to$ **8**.
+ **Reviewer g1NU:** Score raised from **6** $\to$ **8**.

+ We regret that we **did not receive a response** from **Reviewer kbva** during the discussion period to confirm if our revisions resolved their concerns. However, we have conscientiously addressed every issue raised in their initial review through additional experiments, comprehensive clarifications, and new baseline comparisons.

> Key Revisions and Clarifications

We have revised the manuscript to incorporate additional experiments, baselines, and qualitative analyses requested by the reviewers **(highlighted in red for your reference)**. The key improvements are summarized below:

1. Novelty and Comparison to Existing Methods (**Addressing Reviewers btyh, g1NU, kbva**)

+ **Differentiation from ALIGNN & Line Graphs:** We clarified that unlike ALIGNN (which uses line graphs primarily for message passing) or standard edge-feature updates, DeMol treats bonds as **first-class entities** in a dual-graph architecture. We highlighted **our theoretical information-theoretic motivation (Propositions 1-4)** and the specific design of **Double-Helix Blocks** for synergistic fusion, which are absent in prior work.
+ **New Baseline Comparison (ESA):** Per Reviewer kbva's suggestion, we compared DeMol against the "End-to-End Attention-Based Approach" (ESA). DeMol outperforms ESA on QM9 targets, reinforcing our SOTA claims.

2. Qualitative Analysis of Bond Interactions (**Addressing Reviewers UzsJ, g1NU**)

To demonstrate that DeMol explicitly captures complex bond phenomena (resonance and stereoselectivity) rather than just smoothing features, we added extensive visualizations in **Appendix F**:

+ **Attention Maps (Appendix E included in the original version and retained):** We visualized self-attention weights in **original version**, showing that the bond-centric channel captures highly structured, localized geometric patterns (diagonal/off-diagonal blocks), distinct from the global patterns in the atom-centric channel.

+ **t-SNE Visualizations (Appendix F in revised version):** We curated a dataset of stereoisomers (cis- vs. trans-platin) and aromatic systems. DeMol's latent space clearly separates cis/trans isomers and aromatic/non-aromatic rings into four distinct clusters. In contrast, baselines like GEM and LEMON failed to separate the stereoisomers, proving DeMol's superior geometric encoding.

3. Performance Analysis on QM9 (**Addressing Reviewers btyh, kbva**)

We provided a detailed insight into why performance gains are more massive on PCQM4Mv2/OC20 than on QM9. QM9 consists of small, simple molecules (max 9 heavy atoms) where complex bond-level phenomena (long-range resonance, complex stereochemistry) are less prevalent. Despite this, DeMol still achieves **SOTA on 6 out of 12 targets** and the best average rank (2.67), proving robustness even on simpler systems.

4. Technical Details and Efficiency (**Addressing Reviewer UzsJ**)

+ **Inference Time:** We added **Appendix C.1** comparing inference time. DeMol (\~34 ms/molecule) is significantly faster than ensemble approaches like GPS++ (\~100 ms/molecule) while maintaining superior single-model performance.

+ **Pre-training:** We clarified the multi-task pretraining objective (HOMO-LUMO regression + auxiliary geometric tasks) in **Appendix D.1**.

----

We sincerely thank all the reviewers and the Area Chair for their efforts in reviewing our paper. Please let us know if you have any further concerns, and we are willing to answer any further questions you have about our paper. Thank you again for your feedback.



Thanks!



The Authors

---

### Meta-Review · Area_Chair_7ycv · 2025-12-19

**Summary:**

Conventional atom-centric models typically regard chemical bonds merely as pairwise interactions and often overlook complex bonding phenomena such as resonance and stereoselectivity. To address this limitation, the authors propose DeMol, a dual-graph framework whose architecture is motivated by rigorous information-theoretic analysis, demonstrating that additional information can be gained from a bond-centric perspective.

The reviewers raised several issues, including the uniqueness of DeMol’s methodological contribution (particularly in comparison to previous dual-graph and bond-aware graph neural network approaches), the completeness of baseline comparisons, the interpretability and effectiveness of the bond-centric channel, result analysis (especially on small-scale datasets such as QM9), and technical concerns regarding efficiency and pre-training. Although one reviewer (kbva) still has reservations about the novelty and the thoroughness of the empirical analysis, sufficient evidence and follow-up responses to substantiate these concerns were not provided.

**Reviewer Concerns:**

Reviewers focused on the originality of the DeMol method and its differences from existing approaches, particularly requesting clarification on its essential distinctions from dual-graph or line graph methods such as ALIGNN and LEMON. Reviewers also highlighted the importance of demonstrating the model’s ability to capture complex chemical phenomena—such as bond conjugation, aromaticity, and stereoisomerism—suggesting the inclusion of more qualitative analyses (e.g., t-SNE visualizations and attention heatmaps) to intuitively showcase DeMol’s unique advantages. Additionally, some reviewers pointed out that the current baseline comparisons are not sufficiently comprehensive and recommended incorporating newer, more representative methods to strengthen the case for DeMol’s effectiveness. There was also concern about the need for results on inference efficiency, ablation studies, the model’s generalization to molecules of different sizes, and more detailed descriptions of pretraining strategies and their impact. Overall, the reviewers’ feedback centers on the justification of innovation, the comprehensiveness of the experimental design, and the depth of result interpretation.

**Reviewer Scores:**

The initial score was 6642. Prior to the data leak incident, one reviewer responded and raised the score (6642->6662). Although the reviewer with negative score (2) raised concerns regarding experimental comparisons and novelty, they did not provide sufficient evidence to substantiate these criticisms. Therefore, I recommend accepting this paper.

---

### Decision · Program_Chairs · 2026-01-26

Accept (Poster)